# Probable nature of higher-dimensional symmetries underlying mammalian grid-cell activity patterns

Alexander Mathis[1,2]*, Martin B Stemmler[3,4], Andreas VM Herz[3,4]

[1]Department of Molecular and Cellular Biology, Harvard University, Cambridge, United States; [2]Center for Brain Science, Harvard University, Cambridge, United States; [3]Bernstein Center for Computational Neuroscience, Munich, Germany; [4]Fakultät für Biologie, Ludwig-Maximilians-Universität München, Munich, Germany

**Abstract** Lattices abound in nature—from the crystal structure of minerals to the honey-comb organization of ommatidia in the compound eye of insects. These arrangements provide solutions for optimal packings, efficient resource distribution, and cryptographic protocols. Do lattices also play a role in how the brain represents information? We focus on higher-dimensional stimulus domains, with particular emphasis on neural representations of physical space, and derive which neuronal lattice codes maximize spatial resolution. For mammals navigating on a surface, we show that the hexagonal activity patterns of grid cells are optimal. For species that move freely in three dimensions, a face-centered cubic lattice is best. This prediction could be tested experimentally in flying bats, arboreal monkeys, or marine mammals. More generally, our theory suggests that the brain encodes higher-dimensional sensory or cognitive variables with populations of grid-cell-like neurons whose activity patterns exhibit lattice structures at multiple, nested scales.

## Introduction

In mammals, the neural representation of space rests on at least two classes of neurons. 'Place cells' discharge when an animal is near one particular location in its environment (*O'Keefe and Dostrovsky, 1971*). 'Grid cells' are active at multiple locations that span an imaginary hexagonal lattice covering the environment (*Hafting et al., 2005*) and have been found in rats, mice, crawling bats, and human beings (*Hafting et al., 2005*; *Fyhn et al., 2008*; *Yartsev et al., 2011*; *Jacobs et al., 2013*). These cells are believed to build a metric for space.

In these experiments, locomotion occurs on a horizontal plane. Theoretical and numerical studies suggest that the hexagonal lattice structure is best suited for representing such a two-dimensional (2D) space (*Guanella and Verschure, 2007*; *Mathis, 2012*; *Wei et al., 2013*). In general, however, animals move in three dimensions (3D); this is particularly true for birds, tree dwellers, and fish. Their neuronal representation of 3D space may consist of a mosaic of lower-dimensional patches (*Jeffery et al., 2013*), as evidenced by recordings from climbing rats (*Hayman et al., 2011*). Place cells in flying bats, on the other hand, represent 3D space in a uniform and nearly isotropic manner (*Yartsev and Ulanovsky, 2013*).

As mammalian grid cells might represent space differently in 3D than in 2D, we study grid-cell representations in arbitrarily high-dimensional spaces and measure the accuracy of such representations in a population of neurons with periodic tuning curves. We measure the accuracy by the Fisher information (FI). Even though the firing fields between cells overlap, so as to ensure uniform coverage of space, we show how resolving the population's FI can be mapped onto the problem of packing *non-overlapping* spheres, which also plays an important role in other coding problems and

*For correspondence: amathis@
fas.harvard.edu

**Competing interests:** The authors declare that no competing interests exist.

**eLife digest** The brain of a mammal has to store vast amounts of information. The ability of animals to navigate through their environment, for example, depends on a map of the space around them being encoded in the electrical activity of a finite number of neurons. In 2014 the Nobel Prize in Physiology or Medicine was awarded to neuroscientists who had provided insights into this process. Two of the winners had shown that, in experiments on rats, the neurons in a specific region of the brain 'fired' whenever the rat was at any one of a number of points in space. When these points were plotted in two dimensions, they made a grid of interlocking hexagons, thereby providing the rat with a map of its environment.

However, many animals, such as bats and monkeys, navigate in three dimensions rather than two, and it is not clear whether these same hexagonal patterns are also used to represent three-dimensional space. Mathis et al. have now used mathematical analysis to search for the most efficient way for the brain to represent a three-dimensional region of space. This work suggests that the neurons need to fire at points that roughly correspond to the positions that individual oranges take up when they are stacked as tight as possible in a pile. Physicists call this arrangement a face-centered cubic lattice.

At least one group of experimental neuroscientists is currently making measurements on the firing of neurons in freely flying bats, so it should soon be possible to compare the predictions of Mathis et al. with data from experiments.

cryptography (*Shannon, 1948*; *Conway and Sloane, 1992*; *Gray and Neuhoff, 1998*). The optimal lattices are thus the ones with the highest packing ratio—the densest lattices represent space most accurately. This remarkably simple and straightforward answer implies that hexagonal lattices are optimal for representing 2D space. In 3D, our theory makes the experimentally testable prediction that grid cells will have firing fields positioned on a face-centered-cubic lattice or its equally dense non-lattice variant—a hexagonal close packing structure.

Unimodal tuning curves with a single preferred stimulus, which are characteristic for place cells or orientation-selective neurons in visual cortex, have been extensively studied (*Paradiso, 1988*; *Seung and Sompolinsky, 1993*; *Pouget et al., 1999*; *Zhang and Sejnowski, 1999*; *Bethge et al., 2002*; *Eurich and Wilke, 2000*; *Brown and Bäcker, 2006*). This is also true for multimodal tuning curves that are periodic along orthogonal stimulus axes and generate repeating hypercubic (or hyper-rectangular) activation patterns (*Montemurro and Panzeri, 2006*; *Fiete et al., 2008*; *Mathis et al., 2012*). Our results extend these studies by taking more general stimulus symmetries into account and lead us to hypothesize that optimal lattices not only underlie the neural representation of physical space, but will also be found in the representation of other high-dimensional sensory or cognitive spaces.

## Model

### Population coding model for space

We consider the $D$-dimensional space $\mathbb{R}^D$ in which spatial location is denoted by coordinates $x = (x_1, \ldots, x_D) \in \mathbb{R}^D$. The animal's position in this space is encoded by $N$ neurons. The dependence of the mean firing rate of each neuron $i$ on $x$ is called the neuron's tuning curve and will be denoted by $\Omega_i(x)$. To account for the trial-to-trial variability in neuronal firing, spikes are generated stochastically according to a probability $P_i(k_i | \tau \, \Omega_i(x))$ for neuron $i$ to fire $k_i$ spikes within a fixed time window $\tau$. While two neurons can have correlated tuning curves $\Omega_i(x)$, we assume that the trial-to-trial variability of any two neurons is independent of each other. Thus, the conditional probability of the $N$ statistically independent neurons to fire $(k_1, \ldots, k_N)$ spikes at position $x$ summarizes the encoding model:

$$P\big((k_1, \ldots, k_N)\big|x\big) = \prod_{i=1}^{N} P_i\big(k_i\big|\tau \, \Omega_i(x)\big). \tag{1}$$

Decoding relies on inverting this conditional probability by asking: given a spike count vector $K = (k_1, \ldots, k_N)$, where is the animal? Such a position estimate will be written as $\hat{x}(K)$. How precisely the decoding can be done is assessed by calculating the average mean square error of the decoder. The average distance between the real position of the animal $x$ and the estimate $\hat{x}(K)$ is

$$\varepsilon(\hat{x}|x) = \mathbb{E}_{P(K|x)}(||x - \hat{x}(K)||), \tag{2}$$

given the population coding model $P(K|x)$. This error is called the resolution (*Seung and Sompolinsky, 1993*; *Lehmann, 1998*), whereby the term $||.||$ denotes Euclidean distance, $||x|| = \sqrt{\sum_\alpha x_\alpha^2}$. More generally, the covariance matrix $\sum(\hat{x}|x)$ with coefficients $\sum(\hat{x}|x)_{\alpha,\beta} = \mathbb{E}_{P(K|x)}((x_\alpha - \widehat{x_\alpha}(K)) \cdot (x_\beta - \widehat{x_\beta}(K)))$ for spatial dimensions $\alpha, \beta \in \{1, \dots, D\}$, measures the covariance of the different error components, so that the sum of the diagonal elements of $\sum$ is just the resolution $\varepsilon(\hat{x}|x)$. In principle, the resolution depends on both the specific decoder and the population coding model. However, for unbiased estimators, that is, estimators that on average decode the location $x$ as this location $\mathbb{E}_{P(K|x)}(\hat{x}(K)) = x$, the FI provides an analytical measure to assess the highest possible resolution of any such decoder (*Lehmann, 1998*).

## Resolution and Fisher Information

Given a response of $K = (k_1, \dots, k_N)$ spikes across the population, we ask how accurately an ideal observer can decode the stimulus $x$. The FI measures how well one can discriminate nearby stimuli and depends on how $P(x, K)$ changes with $x$. The greater the FI, the higher the resolution, and the lower the error $\varepsilon(\hat{x}|x)$, as these two quantities are inversely related. More precisely, the inverse of the FI matrix $J(x)$,

$$J_{\alpha\beta}(x) = \int \left(\frac{\partial \ln P(K, x)}{\partial x_\alpha}\right)\left(\frac{\partial \ln P(K, x)}{\partial x_\beta}\right) P(K, x)\, dK, \tag{3}$$

bounds the covariance matrix $\sum(\hat{x}|x)$ of the estimated coordinates $x = (x_1, \dots, x_D)$

$$\sum(\hat{x}|x) \geq J(x)^{-1}. \tag{4}$$

The resolution of any unbiased estimator of the encoded stimulus can achieve cannot be greater than $J(x)^{-1}$. This is known as the Cramér-Rao bound (*Lehmann, 1998*). Based on this bound, we will consider the FI as a measure for the resolution of the population code. In particular, we are interested in isotropic and homogeneous representations of space. These two conditions assure that the population has the same resolution at any location and along any spatial axis. Isotropy does not entail that the (global) spatial tuning of an individual neuron, $\Omega_i(x)$, has to be radially symmetric, but merely that the errors are (locally) distributed according to a radially symmetric distribution. For instance, the tuning curve of a grid cell with hexagonal tuning is not radially symmetric around the center of a field (it has three axes), but the posterior is radially symmetric around any given location for a module of such grid cells. Homogeneity requires that the FI $J(x)$ be asymptotically independent of $x$ (as the number of neurons $N$ becomes large); spatial isotropy implies that all diagonal entries in the FI matrix $J(x)$ are equal.

## Periodic tuning curves

Grid cells have periodic tuning curves—they are active at multiple locations, called firing fields, and these firing fields are hexagonally arranged in the environment (*Hafting et al., 2005*). Their periodic structure is given by a hexagonal lattice. The periodic structure of the tuning curve $\Omega_i(x)$ reflects its symmetries, that is, the set of vectors that map the tuning curve onto itself. Since we want to understand how the periodic structure affects the resolution of the population code, we generalize the notion of a grid cell to allow different periodic structures other than just hexagonal. Mathematically, the symmetries of a periodic structure can be described by a lattice $\mathcal{L}$, which is constructed as follows: take a set of independent vectors $(v_\alpha)_{1 \leq \alpha \leq D}$ in $D$-dimensional space $\mathbb{R}^D$, and consider all possible combinations of these vectors and their integer multiples—each such vector combination points to a node of the lattice, such that the union of these represents the lattice itself. For instance, the square lattice (*Figure 1A*, bottom) is given by basis vectors $v_1 = (1, 0)$ and $v_2 = (0, 1)$. Mathematically, the lattice $\mathcal{L} \subset \mathbb{R}^D$ is

$$\mathcal{L} = \sum_{\alpha=1}^{D} k_\alpha v_\alpha \quad \text{for} \quad k_\alpha \in \mathbb{Z},\ v_\alpha \in \mathbb{R}^D, \tag{5}$$

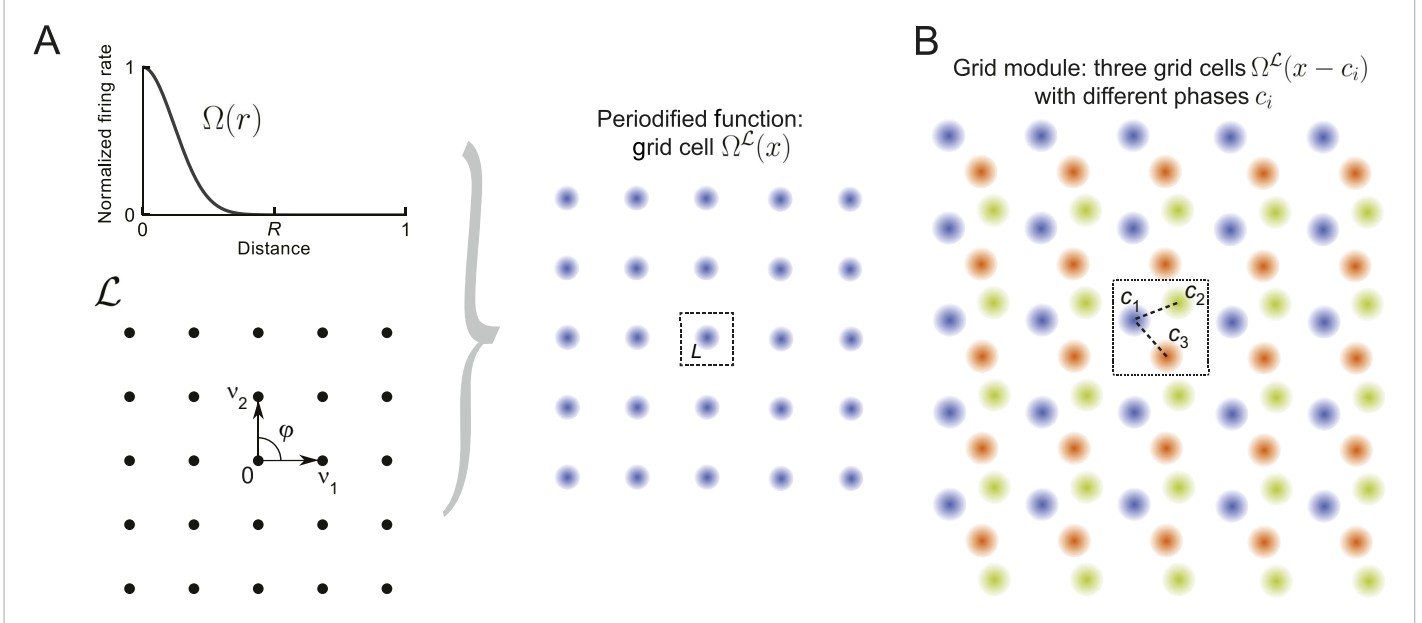

**Figure 1**. Grid cells and modules. (**A**) Construction of a grid cell: Given a tuning shape $\Omega$ and a lattice $\mathcal{L}$, here a square lattice generated by $v_1$ and $v_2$ with $\varphi = \pi/2$, one periodifies $\Omega$ with respect to $\mathcal{L}$. One defines the value of $\Omega^{\mathcal{L}}(x)$ in the fundamental domain $L$ as the value of $\Omega(r)$ applied to the distance from zero and then repeats this map over $\mathbb{R}^2$ like $\mathcal{L}$ tiles the space. This construction can be used for lattices $\mathcal{L}$ of arbitrary dimensions (**Equation 7**). (**B**) Grid module: The firing rates of three grid cells (orange, green, and blue) are indicated by color intensity. The cells' tuning is identical ($\Omega$ and $\mathcal{L}$ are the same), yet they differ in their spatial phases $c_i$. Together, such identically tuned cells with different spatial phases define a grid module.

for which $(v_\alpha)_{1 \leq \alpha \leq D}$ is a basis of $\mathbb{R}^D$. We will not consider degenerate lattices. In this work, we follow the nomenclature from **Conway and Sloane (1992)**. Applied fields might differ slightly in their terminology, especially regarding naming conventions for packings, which are generalizations of lattices (**Whittaker, 1981**; **Nelson, 2002**). We will address these generalizations of lattices below.

Based on such a lattice $\mathcal{L}$, we construct periodic tuning curves as illustrated in **Figure 1A**. We start with a lattice $\mathcal{L}$ and a tuning shape $\Omega : \mathbb{R}^+ \to [0, 1]$ that decays from unity to zero; $\Omega(r)$ describes the firing rate of the periodified tuning curve at distance $r$ from any lattice point and should be at least twice continuously differentiable. Each lattice point $p \in \mathcal{L}$ has a domain $V_p \subset \mathbb{R}^D$ called the Voronoi region, which is defined as

$$V_p = \left\{ x \in \mathbb{R}^D \mid ||x - p|| < ||x - q|| \ \forall q \in \mathcal{L} \wedge p \neq q \right\}, \tag{6}$$

that contains all points $x$ that are closer to $p$ than to any other lattice point $q$. Note that $V_p \cap V_q = \phi$ if $p \neq q$ and that for all $p, q \in \mathcal{L}$ there exists a unique vector $v \in \mathcal{L}$ with $V_p = V_q + v$.

The domain that contains the null (0) vector is called the fundamental domain and is denoted by $L := V_0$. For each $x \in \mathbb{R}^D$ there is a unique lattice point $p \in \mathcal{L}$ that maps $x$ into the fundamental domain: $x - p \in L$. Let us call this mapping $\pi_{\mathcal{L}}$. With this notation one can periodify $\Omega$ onto $\mathcal{L}$ by defining a grid cell's tuning curve as $\Omega^{\mathcal{L}}$:

$$\Omega^{\mathcal{L}}(x) : \mathbb{R}^D \to \mathbb{R}^+, \ x \mapsto f_{max} \cdot \Omega\left( ||\pi_{\mathcal{L}}(x)||^2 \right), \tag{7}$$

where $f_{max}$ is the peak firing rate of the neuron. Note that throughout the paper we set $f_{max} = \tau = 1$, for simplicity. As illustrated in **Figure 1A**, within the fundamental domain $L$, the tuning curve $\Omega^{\mathcal{L}}$ defined above is radially symmetric. This pattern is repeated along the nodes of $\mathcal{L}$, akin to ceramic tiling.

A grid module is defined as an ensemble of $M$ grid cells $\Omega_i^{\mathcal{L}}$, $i \in \{1, \ldots, M\}$ with identical, but spatially shifted tuning curves, that is, $\Omega_i^{\mathcal{L}}(x) = \Omega^{\mathcal{L} + c_i}(x)$ and spatial phases $c_i \in L$ (see **Figure 1B**). The various phases within a module can be summarized by their phase density $\rho(c) = \sum_{i=1}^{M} \delta(c - c_i)$. This definition is motivated by the observation of spatially shifted hexagonally tuned grid cells in the entorhinal cortex of rats (**Hafting et al., 2005**; **Stensola et al., 2012**).

Any grid module is uniquely characterized by its signature $(\Omega, \rho, \mathcal{L})$. To investigate the role of different periodic structures, we can fix the tuning shape $\Omega$ and density $\rho$ and solely vary the lattice $\mathcal{L}$ to find the lattice that yields the highest FI.

## Results

To determine how the resolution of a grid module depends on the periodic structure $\mathcal{L}$, we compute the population FI $J_\varsigma(x)$ for a module of grid cells with signature $\varsigma = (\Omega, \rho, \mathcal{L})$, which describes the tuning shape, the density of firing fields, and the lattice. By fixing the tuning shape $\Omega$ and the number $|\rho| = M$ of spatial phases, we can compare the resolution for different periodic structures. (**Table 1** contains a glossary of the variables.)

### Scaling of lattices and nested grid codes

Our grid-cell construction has one obvious degree of freedom, the length scale or grid size of the lattice $\mathcal{L}$, that is, the width of the fundamental domain $L$. For a module with signature $\varsigma = (\Omega, \rho, \mathcal{L})$ and for arbitrary scaling factor $\lambda > 0$, the rescaled construction $\lambda\varsigma := (\Omega(\lambda r), \rho(\lambda x), \lambda \cdot \mathcal{L})$ is a grid module too. The corresponding tuning curve satisfies $(\Omega \circ \lambda)_{\lambda\mathcal{L}}(x) = \Omega_{\mathcal{L}}(\lambda x)$ and is thus merely a scaled version of the former. Indeed, as we show in the 'Material and methods' section, the FI of the rescaled module is $\lambda^{-2} J_\varsigma(0)$. The Cramér-Rao bound (**Equation 4**) implies that the local resolution of an unbiased estimator could thus rapidly improve with a finer grid size, that is, decreasing $\lambda$.

However, for any grid module $\varsigma = (\Omega, \rho, \mathcal{L})$ the posterior probability, that is, the likelihood of possible positions given a particular spike count vector $K = (k_1, \ldots, k_N)$, is also periodic. This follows from Bayes rule:

$$P(x|K) = \frac{P(K|x) \cdot P(x)}{P(K)} \propto P(x) \prod_{i=1}^{N} P_i\left(k_i | \tau \, \Omega_i^{\mathcal{L}}(x)\right). \tag{8}$$

**Table 1.** List of acronyms, variables, and terms

| | |
|---|---|
| $D$ | Dimension of the stimulus space $\mathbb{R}^D$ |
| FI | Fisher information, usually denoted by $J$ (**Equation 3**) |
| $\mathcal{L}$ | Non-degenerate point lattice describing periodic structure (**Equation 5**) |
| $L$ | Fundamental domain of $\mathcal{L}$, which is the Voronoi cell containing 0 (**Equation 6**) |
| $\Omega$ | Tuning shape |
| $\text{supp}(\Omega)$ | Support of $\Omega$, that is, the subset where $\Omega$ does not vanish |
| $\Omega^{\mathcal{L}}$ | Periodified tuning curve on $\mathbb{R}^D$, where $\mathcal{L}$ is a $D$-dimensional lattice and $\Omega$ a tuning curve. Simply referred to as a 'grid cell' (**Equation 7**) |
| $\rho$ | Phase density of grid cells' phases $c_i$ within a module $\rho(c) = \sum_{i=1}^{M} \delta(c - c_i)$ |
| $M$ | Number of phases in grid module $\int_L \rho = M$ |
| $\varsigma = (\Omega, \rho, \mathcal{L})$ | Signature defining a grid module, which is an ensemble of grid cells differing in spatial phases $c_i$, defined by $\rho$ and tuning curves given by $\Omega^{\mathcal{L}}$ |
| $\det(\mathcal{L})$ | Determinant of lattice $\mathcal{L}$ (equal to volume of $L$) |
| $B_R(0)$ | Subset of $\mathbb{R}^D$ containing all points with distance less than $R$ from 0 |
| $\Delta(\mathcal{L})$ | Packing ratio of a lattice, that is, the volume of the largest $B_R(0)$ that fits inside $L$ divided by $\det(\mathcal{L})$ (**Equation 15**) |
| $\mathcal{H}, \mathcal{Q}$ | Hexagonal and square planar lattice of unit node-to-node distance (**Figure 2**) |
| $\mathcal{FCC}, \mathcal{BCC}, \mathcal{C}$ | Face-centered, body-centered, and cubic lattice of unit node-to-node distance, respectively (**Figure 4**). |
| $\text{tr}J$ | Trace of the FI, that is, the sum of diagonal elements |
| $J_\varsigma$ | Population FI of grid module with signature $\varsigma$ |
| $\text{tr}J_{\mathcal{L}}, \text{tr}J_{\mathcal{Q}}, \text{tr}J_{\mathcal{H}}$ | Trace of FI per neuron for lattice $\mathcal{L}$ ($\mathcal{Q}$ and $\mathcal{H}$, respectively) with fixed bump-like $\Omega$ defined in **Equation 26** |
| $\text{tr}J_{\mathcal{L}}^M$ | Trace of FI for lattice $\mathcal{L}$ for $M$ randomly distributed phases in $L$ for the same bump function |

Since the right hand side is invariant under operations of $\mathcal{L}$ on x, so is the left hand side of this equation. Thus, the multiple firing fields of a grid cell cannot be distinguished by a decoder, so that for $\lambda \to 0$ the global resolution approaches the a priori uncertainty (*Mathis et al., 2012a*, *2012b*). By combining multiple grid modules with different spatial periods one can overcome this fundamental limitation, counteracting the ambiguity caused by periodicity and still preserving the highest resolution at the smallest scale. Thus, one arrives at nested populations of grid modules, whose spatial periods range from coarse to fine. The FI for an individual module at one scale determines the optimal length scale of the next module (*Mathis et al., 2012a*, *2012b*). The larger the FI per module, the greater the refinement at subsequent scales can be (*Mathis et al., 2012a*, *2012b*). This result emphasizes the importance of finding the lattice that endows a grid module with maximal FI, but also highlights that the specific scale of the lattices can be fixed for this study.

## FI of a grid module with lattice $\mathcal{L}$

We now calculate the FI for a grid module with signature $\varsigma = (\Omega, \rho, \mathcal{L})$. For cells whose firing is statistically independent (*Equation 1*), the joint probability factorizes; therefore, the population FI is just the sum over the individual FI contributions by each neuron, $\boldsymbol{J}_\varsigma(x) = \sum_{i=1}^{M} \boldsymbol{J}_{\Omega_i^{\mathcal{L}}}(x)$. The individual neurons only differ by their spatial phase $c_i$, thus $\boldsymbol{J}_{\Omega_i^{\mathcal{L}}}(x) = \boldsymbol{J}_{\Omega^{\mathcal{L}}}(x - c_i)$. Consequently, $\boldsymbol{J}_\varsigma(x) = \sum_{i=1}^{M} \boldsymbol{J}_{\Omega^{\mathcal{L}}}(x - c_i)$, depends only on the function $\boldsymbol{J}_{\Omega^{\mathcal{L}}}(r)$ and the deviations $x - c_i$, where $c_i$ is the closest lattice point of $c_i + \mathcal{L}$ to x. If the grid-cell density $\rho$ is uniform across $\mathcal{L}$, then for all $x \in \mathbb{R}^D$: $\boldsymbol{J}_\varsigma(x) \approx \boldsymbol{J}_\varsigma(0)$. It therefore suffices to only consider the FI at the origin, which can be written as:

$$\boldsymbol{J}_\varsigma(0) = \sum_{i=1}^{M} \boldsymbol{J}_{\Omega^{\mathcal{L}}}(c_i) = \int_L \boldsymbol{J}_{\Omega^{\mathcal{L}}}(c)\rho(c)\mathrm{d}c. \tag{9}$$

For uniformly distributed spatial phases $c_i$ and increasing number of neurons $M$, the law of large numbers implies

$$\lim_{M \to \infty} \left| \frac{\det(\mathcal{L})}{M} \boldsymbol{J}_\varsigma(0) - \int_L \boldsymbol{J}_{\Omega^{\mathcal{L}}}(c) \, \mathrm{d}c \right| = 0. \tag{10}$$

Here, $\det(\mathcal{L})$ denotes the volume of the fundamental domain. Thus, for large numbers of neurons $M = \int_L \rho(c)\mathrm{d}c$ we obtain

$$\boldsymbol{J}_\varsigma(0) \approx \frac{M}{\det(\mathcal{L})} \int_L \boldsymbol{J}_{\Omega^{\mathcal{L}}}(c)\mathrm{d}c. \tag{11}$$

This means that the population FI at 0 is approximately given by the average FI within the fundamental domain $L$ times the number of neurons $M$. Let us now assume that supp($\Omega$) = [0, R] for some positive radius $R$. Outside of this radius, the tuning shape is zero and the firing rate vanishes. So the spatial phases of grid cells that contribute to the FI at $x = 0$ lie within the ball $B_R(0)$. If we now also assume that this ball is contained in the fundamental domain, $B_R(0) \subset L$, we get

$$\int_L \boldsymbol{J}_{\Omega^{\mathcal{L}}}(c)\mathrm{d}c = \int_{B_R(0)} \boldsymbol{J}_{\Omega^{\mathcal{L}}}(c)\mathrm{d}c. \tag{12}$$

This result implies that any grid code $\varsigma = (\Omega, \rho, \mathcal{L})$, with large $M$, supp($\Omega$) = [0, R], and $B_R(0) \subset L$, satisfies

$$\boldsymbol{J}_\varsigma(0) \approx \frac{M}{\det(\mathcal{L})} \int_{B_R(0)} \boldsymbol{J}_{\Omega^{\mathcal{L}}}(c)\mathrm{d}c. \tag{13}$$

The FI at the origin is therefore approximately equal to the product of the mean FI contribution of cells within a $R$-ball around 0 and the number of neurons $M$, weighted by the ratio of the volume of the $R$-ball to the area of the fundamental domain $L$. Due to the radial symmetry of $\Omega^{\mathcal{L}}$, the FI matrix $\boldsymbol{J}_{\Omega^{\mathcal{L}}}(c)$ is diagonal with identical entries, guaranteeing the spatial resolution's isotropy. The error for each coordinate axis is bounded by the same value, that is, the inverse of the diagonal element $1/\boldsymbol{J}_\varsigma(0)_{ii}$, for such a population. Instead of considering the FI matrix $\boldsymbol{J}_\varsigma(0)$, we can therefore consider the trace of $\boldsymbol{J}_\varsigma(0)$, which is the sum over the diagonal of $\boldsymbol{J}_\varsigma(0)$. According to *Equation 4*, $1/\mathrm{tr}\boldsymbol{J}_\varsigma(0)$ bounds the mean square error summed across all dimensions $\varepsilon(\hat{x}|x)$.

For two lattices $\mathcal{L}_1, \mathcal{L}_2$, with $B_R(0) \subset L_1 \cap L_2$ we consequently obtain

$$\frac{\operatorname{tr} J_{\Omega^{\mathcal{L}_1}}}{\operatorname{tr} J_{\Omega^{\mathcal{L}_2}}} = \frac{\det(\mathcal{L}_2)}{\det(\mathcal{L}_1)}, \tag{14}$$

which signifies that the resolution of the grid module is inversely proportional to the volumes of their fundamental domains. The periodic structure $\mathcal{L}$ thus has a direct impact on the resolution of the grid module. This result implies that finding the maximum FI translates directly into finding the lattice with the highest packing ratio.

## Packing ratio of lattices

The sphere packing problem is of general interest in mathematics (*Conway and Sloane, 1992*) and has wide-ranging applications from crystallography to information theory (*Barlow, 1883*; *Shannon, 1948*; *Whittaker, 1981*; *Gray and Neuhoff, 1998*; *Gruber, 2004*). When packing $R$-balls $B_R$ in $\mathbb{R}^D$ in a non-overlapping fashion, the density of the packing is defined as the fraction of the space covered by balls. For a lattice $\mathcal{L}$, it is given by

$$\frac{\operatorname{vol}(B_R(0))}{\det(\mathcal{L})}, \tag{15}$$

which is known as the packing ratio $\Delta(\mathcal{L})$ of the lattice. For a given lattice, this ratio is maximized by choosing the largest possible $R$, known as the packing radius, which is defined as the in-radius of a Voronoi region containing the origin (*Conway and Sloane, 1992*). *Figure 2* depicts the disks with the largest in-radius for the hexagonal and the square lattice in blue and illustrates the packing ratio.

## FI and packing ratio

We now come to the main finding of this study: among grid modules with different lattices, the lattice with the highest packing ratio leads to the highest spatial resolution.

To derive this result, let us fix a tuning shape $\Omega$ with $\operatorname{supp}(\Omega) = [0, R]$, lattices $\mathcal{L}_j$ such that $B_R(0) \subset L_j$ for $1 \leq j \leq K$, and uniform densities $\rho$ for each fundamental domain of equal cardinality $M$. Any linear order on the packing ratios,

$$\Delta(\mathcal{L}_1) \leq \ldots \leq \Delta(\mathcal{L}_j) \leq \ldots \leq \Delta(\mathcal{L}_K), \tag{16}$$

is translated by *Equation 14* into the same order for the traces of the FI

$$\operatorname{tr} J_{\Omega^{\mathcal{L}_1}} \leq \ldots \leq \operatorname{tr} J_{\Omega^{\mathcal{L}_j}} \leq \ldots \leq \operatorname{tr} J_{\Omega^{\mathcal{L}_K}}, \tag{17}$$

and thus the resolution of these modules: the higher the packing ratio, the higher the FI of a grid module.

The condition $\operatorname{supp}(\Omega) = [0, R]$ with $B_R(0) \subset L$, although restrictive, is consistent with experimental observations that grid cells tend to stop firing between grid fields and that the typical ratio between field radius and spatial period is well below 1/2 (*Hafting et al., 2005*; *Brun et al., 2008*; *Giocomo et al., 2011*). Generally, the tuning width that maximizes the FI does not necessarily satisfy this condition; see *Figures 3, 4*, in which the optimal support radius of the tuning curve $\theta_2$ is greater than the in-radius $R = 1/2$ of $L$. The same observation will hold in higher dimensions ($D > 2$), consistent with the finding that the optimal tuning width for Gaussian tuning curves increases with the number of spatial dimensions, whether space is infinite (*Zhang and Sejnowski, 1999*) or finite (*Brown and Bäcker, 2006*). When the radius R of the support of the tuning curve exceeds the in-radius, the optimal lattice can be *different* from the densest one as we will show numerically for specific tuning curves and Poisson noise. However, with well separated fields, like those observed experimentally, the densest lattice provides the highest resolution for any tuning shape $\Omega$, as we just demonstrated.

The optimal packing ratio of lattices for low-dimensional space is well known. Having established our main result, we can now draw on a rich body of literature, in particular *Conway and Sloane (1992)*, to discuss the expected firing-field structure of grid cells in 2D and 3D environments.

## Optimal 2D grid cells

With a packing ratio of $\pi/\sqrt{12}$, the hexagonal lattice is the densest lattice in the plane (*Lagrange, 1773*). According to *Equation 14*, the hexagonal lattice is the optimal arrangement for grid-cell firing

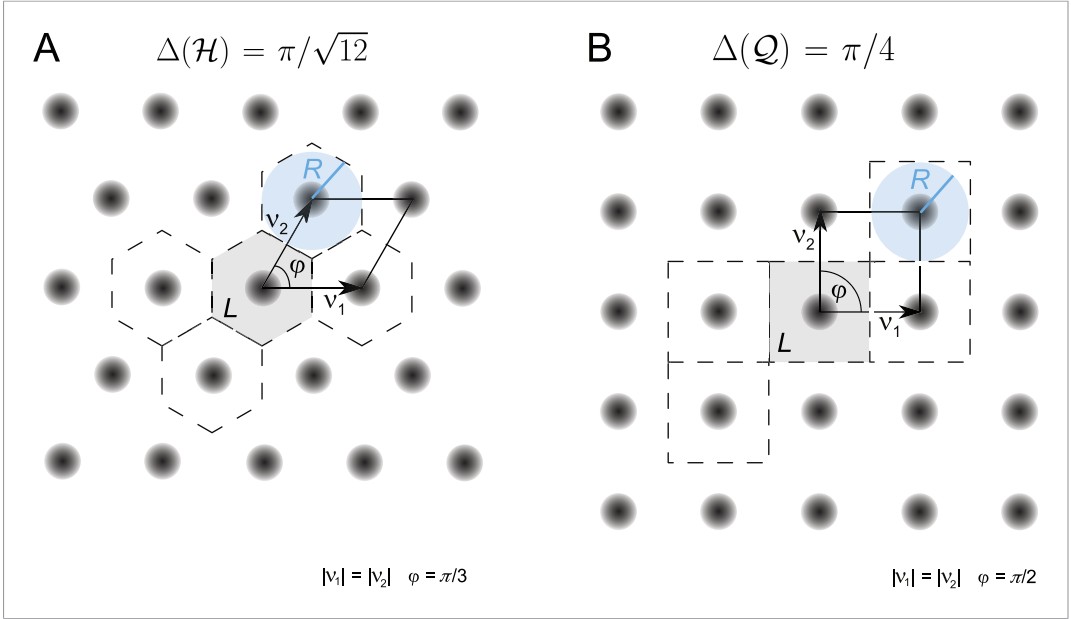

**Figure 2.** Periodified grid-cell tuning curve $\Omega^{\mathcal{L}}$ for two planar lattices, (**A**) the hexagonal (equilateral triangle) lattice $\mathcal{H}$ and (**B**) the square lattice $\mathcal{Q}$, together with the basis vectors $v_1$ and $v_2$. These are $\pi/3$ apart for the hexagonal lattice and $\pi/2$ for the square lattice. The fundamental domain, that is, the Voronoi cell around 0, is shown in gray. A few other domains that have been generated according to the lattice symmetries are marked by dashed lines. The blue disk shows the disk with maximal radius $R$ that can be inscribed in the two fundamental domains. For equal and unitary node-to-node distances, that is, $|v_1| = |v_2| = 1$, the maximal radius equals 1/2 for both lattices. The packing ratio $\Delta$ is $\Delta(\mathcal{H}) = \pi/\sqrt{12}$ for the hexagonal and $\Delta(\mathcal{Q}) = \pi/4$ for the square lattice; the hexagonal lattice is approximately 15.5% denser than the square lattice.

fields on the plane. For example, it outperforms the quadratic lattice, which has a density of $\pi/4$, by about 15.5% (see **Figure 2**). Consequently, the FI of a grid module periodified along a hexagonal lattice outperforms one periodified along a square lattice by the same factor.

To provide a tangible example, we calculated the trace of the average FI per neuron $\mathrm{tr}J_\varsigma / \int_L \rho$ for signature $\varsigma = (\Omega, \rho, \mathcal{L})$ and chose the lattice $\mathcal{L}$ to either be the hexagonal lattice $\mathcal{H}$ or the quadratic lattice $\mathcal{Q}$. We denote the trace of the average FI per neuron as: $\mathrm{tr}J_{\mathcal{L}} = \mathrm{tr}J_\varsigma / \int_L \rho$; $\mathrm{tr}J_{\mathcal{H}}$ and $\mathrm{tr}J_{\mathcal{Q}}$ are similarly defined. We considered Poisson spike statistics and used a bump-like tuning shape $\Omega$ (**Equation 26**, 'Materials and methods' section). The tuning shape $\Omega$ depends on two parameters $\theta_1$ and $\theta_2$, where $\theta_1$ controls the slope of the flank in $\Omega$ and $\theta_2$ defines the support radius. The periodified tuning curve $\Omega^{\mathcal{Q}}$ is illustrated for different parameters in the top of **Figure 3A** and in **Figure 3—figure supplement 1**.

**Figure 3A** depicts $\mathrm{tr}J_{\mathcal{H}}$ and $\mathrm{tr}J_{\mathcal{Q}}$ for various values of $\theta_1$ and $\theta_2$. Quite generally, the FI is larger for grid modules with broad tuning (large $\theta_2$) and steep tuning slopes (small $\theta_1$). **Figure 3A** also demonstrates that as long as $\theta_2 \leq 1/2$, $\mathrm{tr}J_{\mathcal{H}}$ consistently outperforms $\mathrm{tr}J_{\mathcal{Q}}$. But how large is this effect? As predicted by our theory, the grid module with the hexagonal lattice outperforms the square lattice by the relation of packing ratios $\sqrt{3}/2$, as long as the support radius $\theta_2$ is within the fundamental domain of the hexagonal and the square lattice of unit length, that is, $\theta_2 \leq 1/2$ (bottom of **Figure 3A**). As the support radius becomes larger, the FI of the hexagonal lattice is no longer necessarily greater than that of the square lattice; the specific interplay of tuning curve and boundary shape determines which lattice is better: for $\theta_1 = 1/4$, $\mathrm{tr}J_{\mathcal{H}}/\mathrm{tr}J_{\mathcal{Q}}$ drops quickly beyond $\theta_2 = 0.5$, even though, for $\theta_1 = 1$, the ratio stays constant up to $\theta_2 = 0.6$.

Next we calculated the FI per neuron for a larger family of planar lattices generated by two unitary basis vectors with angle $\varphi$. **Figure 3B** displays $\mathrm{tr}J_{\mathcal{L}}$ for $\varphi \in [\pi/3, \pi/2]$, slope parameter $\theta_1 = 1/4$, and different support radii $\theta_2$. For the lattice to have unitary length, the value $\varphi$ cannot go below $\pi/3$. The $\mathrm{tr}J_{\mathcal{L}}$ decays with increasing angle $\varphi$. Indeed, according to **Equation 13**, the FI falls like $1/\det \mathcal{L} = 1/\sin(\varphi)$ so that the maximum is achieved for the hexagonal lattice with $\pi/3$.

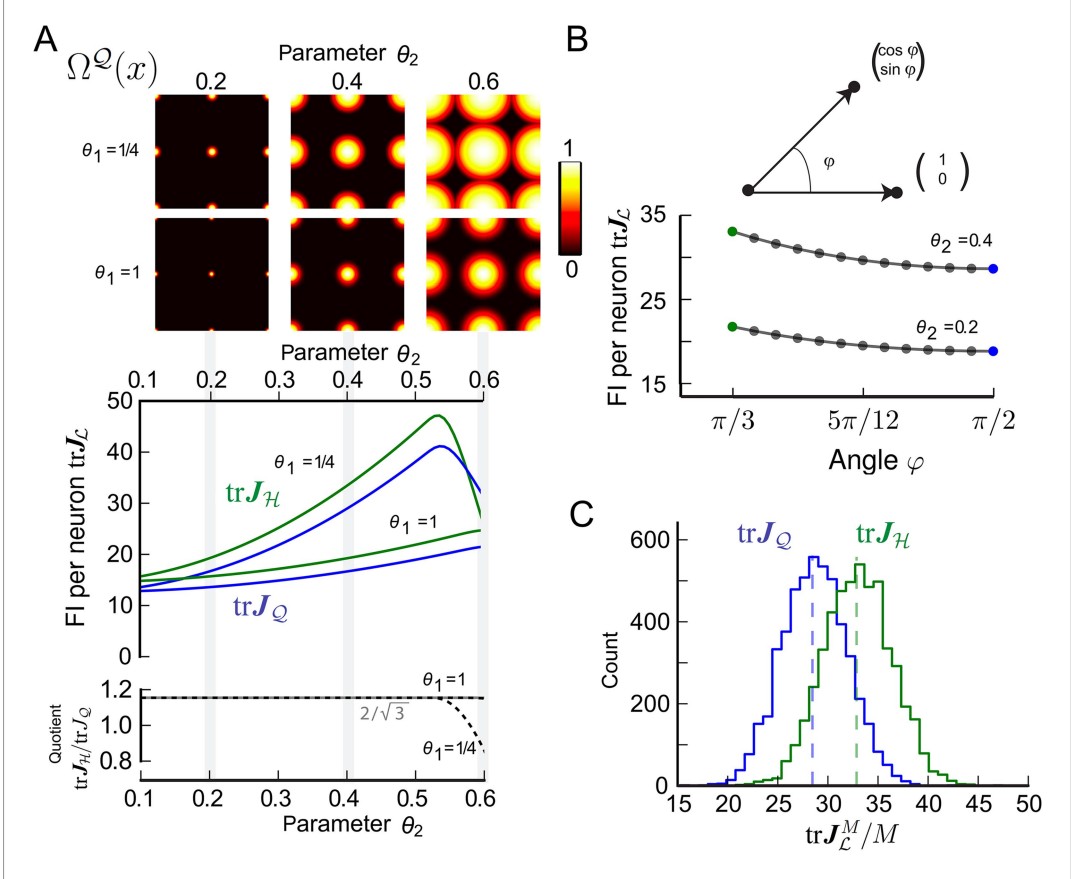

**Figure 3.** Fisher information for modules of two-dimensional grid cells. (**A**) Top: Periodified bump-function $\Omega$ and square lattice $\mathcal{L}$, for various parameter combinations $\theta_1$ and $\theta_2$. Here, $\theta_1$ modulates the decay and $\theta_2$ the support. Middle: Average trace $\mathrm{tr}\mathbf{J}_{\mathcal{L}}$ of the Fisher information (FI) for uniformly distributed grid cells $\Omega^{\mathcal{L}}$. Hexagonal ($\mathcal{H}$) and square ($\mathcal{Q}$) lattices are considered for different $\theta_1$ and $\theta_2$ values. The FI of the hexagonal grid cells outperforms the quadratic grid when support is fully within the fundamental domain ($\theta_2 < 0.5$, see main text). Bottom: Ratio $\mathrm{tr}\mathbf{J}_{\mathcal{H}}/\mathrm{tr}\mathbf{J}_{\mathcal{Q}}$ as a function of the tuning parameter $\theta_2$. For $\theta_2 < 0.5$, the hexagonal population offers 3/2 times the resolution of the square population, as predicted by the respective packing ratios. (**B**) Average $\mathrm{tr}\mathbf{J}_{\mathcal{L}}$ for grid cells distributed uniformly in lattices generated by basis vectors separated by an angle $\varphi$ (basis depicted above graph). $\mathrm{tr}\mathbf{J}_{\mathcal{L}}$ behaves like $1/\sin(\varphi)$ and has its maximum at $\pi/3$. (**C**) Distribution of 5000 realizations of $\mathrm{tr}\mathbf{J}_{\mathcal{L}}^{M}/M$ at 0 for a population of $M = 200$ randomly distributed neurons. For both the hexagonal and square lattice, parameters are $\theta_1 = 1/4$ and $\theta_2 = 0.4$. The means closely match the average values in (**A**). However, due to the finite neuron number the FI varies strongly for different realizations, and in about 20% of the cases a square lattice module outperforms a hexagonal lattice.

The following figure supplement is available for figure 3:

**Figure supplement 1**. The firing rate and Fisher information of the bump tuning shape.

The FIs $\mathrm{tr}\mathbf{J}_{\mathcal{L}}$ are averages over all phases, under the assumption that the density of phases tends to a constant; but are these values also indicative for small neural populations? To answer this question, we calculated the FI for populations with 200 neurons, as some putative grid cells are found in patches of this size (**Ray et al., 2014**). For $M = 200$ randomly chosen phases (**Figure 3C**), the mean of the normalized FI $\mathrm{tr}\mathbf{J}_{\mathcal{L}}^{M}/M$ over 5000 realizations is well captured by the FI per neuron calculated in **Figure 3A**. Because of fluctuations in the FI, however, the square lattice is better than the hexagonal lattice in about 20% of the cases.

Our theory implies that for radially symmetric tuning curves the hexagonal lattice provides the best resolution among all planar lattices. This conclusion agrees with earlier findings: Wei et al. considered a notion of resolution defined as the range of the population code per smallest distinguishable scale

and then demonstrated that a population of nested grid cells with hexagonal tuning is optimal for a winner-take-all and Bayesian maximum likelihood decoders (*Wei et al., 2013*). Guanella and Verschure numerically compared hexagonal to other regular lattices based on maximum likelihood decoding (*Guanella and Verschure, 2007*).

## Optimal lattices for 3D grid cells

Gauss proved that the packing ratio of any cubic lattice is bounded by $\pi/(3\sqrt{2})$ and that this value is attained for the face-centered cubic ($\mathcal{FCC}$) lattice (*Gauss, 1831*) illustrated in *Figure 4A*. This implies that the optimal 3D grid-cell tuning is given by the $\mathcal{FCC}$ lattice. For comparison, we also calculated the average population FI for two other important 3D lattices: the cubic lattice ($\mathcal{C}$) and the body-centered cubic lattice ($\mathcal{BCC}$), both shown in *Figure 4A*.

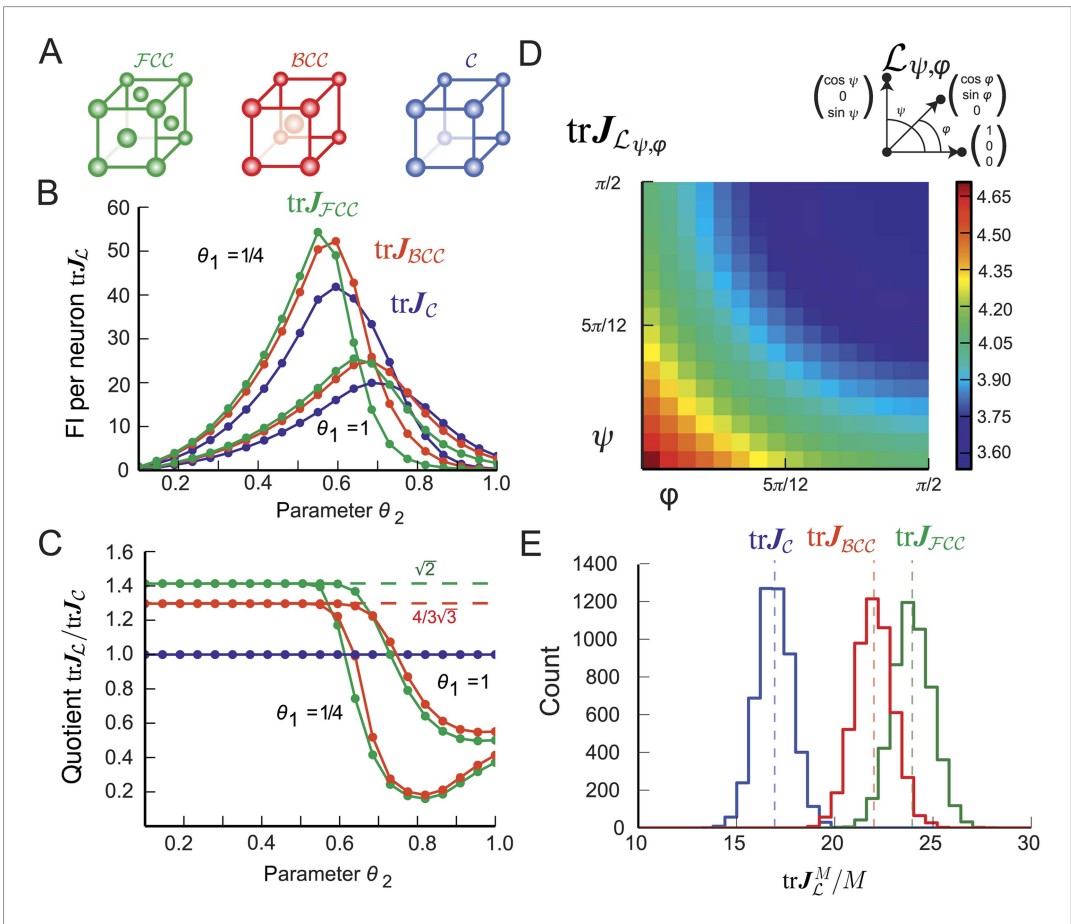

**Figure 4**. Fisher information for modules of 3D grid cells. (**A**) The three lattices considered: face-centered cubic ($\mathcal{FCC}$), body-centered cubic ($\mathcal{BCC}$), and cubic ($\mathcal{C}$). (**B**) $\mathrm{tr}\mathbf{J}_{\mathcal{L}}$ for the periodified bump-function $\Omega$ for the three lattices and various parameter combinations $\theta_1$ and $\theta_2$. The Fisher information (FI) of the $\mathcal{FCC}$ grid cells outperforms the other lattices when the support is fully within the fundamental domain ($\theta_2 < 0.5$, see main text). For larger $\theta_2$ the best lattice depends on the relation between the Voronoi cell's boundary and the tuning curve. (**C**) Ratio $\mathrm{tr}\mathbf{J}_{\mathcal{L}}/\mathrm{tr}\mathbf{J}_{\mathcal{C}}$ as a function of $\theta_2$ for $\mathcal{L} \in \{\mathcal{FCC}, \mathcal{BCC}, \mathcal{C}\}$. For $\theta_2 < 0.5$, the hexagonal population has 3/2 times the resolution of the square population, as predicted by the packing ratios. (**D**) Average $\mathrm{tr}\mathbf{J}_{\mathcal{L}_{\varphi,\psi}}$ for uniformly distributed grid cells within a lattice $\mathcal{L}_{\varphi,\psi}$ generated by basis vectors separated by angles $\varphi$ and $\psi$ (as shown above; $\theta_1 = \theta_2 = 1/4$). $\mathrm{tr}\mathbf{J}_{\mathcal{L}_{\varphi,\psi}}$ behaves like $1/(\sin\varphi\cdot\sin\psi)$ and has its maximum for the lattice with the smallest volume. (**E**) Distribution of 5000 realizations of $\mathrm{tr}\mathbf{J}_{\mathcal{L}}^{M}/M$ at 0 for a population of $M = 200$ randomly distributed neurons. Parameters: $\theta_1 = 1/4$, $\theta_2 = 0.4$. The means closely match the averages in (**B**). Due to the finite neuron number, the FI varies strongly for different realizations.

Keeping the bump-like tuning shape $\Omega$ and independent Poisson noise, we compared the resolution of grid modules with such lattices (*Figure 4B*). Their averaged trace of FI is denoted by $\mathrm{tr}J_{\mathcal{FCC}}$, $\mathrm{tr}J_{\mathcal{BCC}}$, and $\mathrm{tr}J_{\mathcal{C}}$, respectively. As long as the support $\theta_2$ of $\Omega$ is smaller than 1/2, the support is a subset of the fundamental domain of all three lattices. Hence, the trace of the population FI of the $\mathcal{FCC}$ outperforms both the $\mathcal{BCC}$ and $\mathcal{C}$ lattices. As the ratios of the trace of the population FI scales with the packing ratio (*Figure 4C*), $\mathcal{FCC}$-grid cells provide roughly 41% more resolution for the same number of neurons than do $\mathcal{C}$-grid cells. Similarly, $\mathcal{FCC}$-grid cells provide 8.8% more FI than $\mathcal{BCC}$-grid cells.

Next we calculated the FI per neuron for a large family of cubic lattices $\mathcal{L}_{\varphi,\psi}$ generated by three unitary basis vectors with spanning angles $\varphi$ and $\psi$. *Figure 4D* displays $\mathrm{tr}J_{\mathcal{L}_{\varphi,\psi}}$ for $\theta_1 = \theta_2 = 1/4$ and various $\varphi$ and $\psi$. The resolution $\mathrm{tr}J_{\mathcal{L}}$ decays with increasing angles and has its maximum for the lattice with the smallest volume as predicted by *Equation 13*.

To study finite-size effects, we simulated 5000 populations of 200 grid cells with random spatial phases. Qualitatively, the results (*Figure 4E*) match those in 2D (*Figure 3C*). Despite the small module size, $\mathcal{FCC}$ outperformed the cubic lattice $\mathcal{C}$ in all simulated realizations.

## Equally optimal non-lattice solutions for grid-cell tuning

Fruit is often arranged in an $\mathcal{FCC}$ formation (*Figure 5A*). One arrives at this lattice by starting from a layer of hexagonally placed spheres. This requires two basis vectors to be specified and is the densest packing in 2D. To maximize the packing ratio in 3D, the next layer of hexagonally arranged spheres has to be stacked as tightly as possible. There are two choices for the third and final basis vector achieve this packing, denoted as $\gamma_1$ and $\gamma_2$ in *Figure 5B* (modulo hexagonal symmetry). If one chooses $\gamma_1$, then two layers below there is no sphere with its center at location $\gamma_1$, but instead there is one at $\gamma_2$ (and vice versa). This stacking of layers is shown in *Figure 5C* and generates the $\mathcal{FCC}$ lattice.

One could achieve the same density by choosing $\gamma_1$ for both the top layer and the layer below the basis layer. Yet as this arrangement, called hexagonal close packing ($\mathcal{HCP}$), cannot be described by

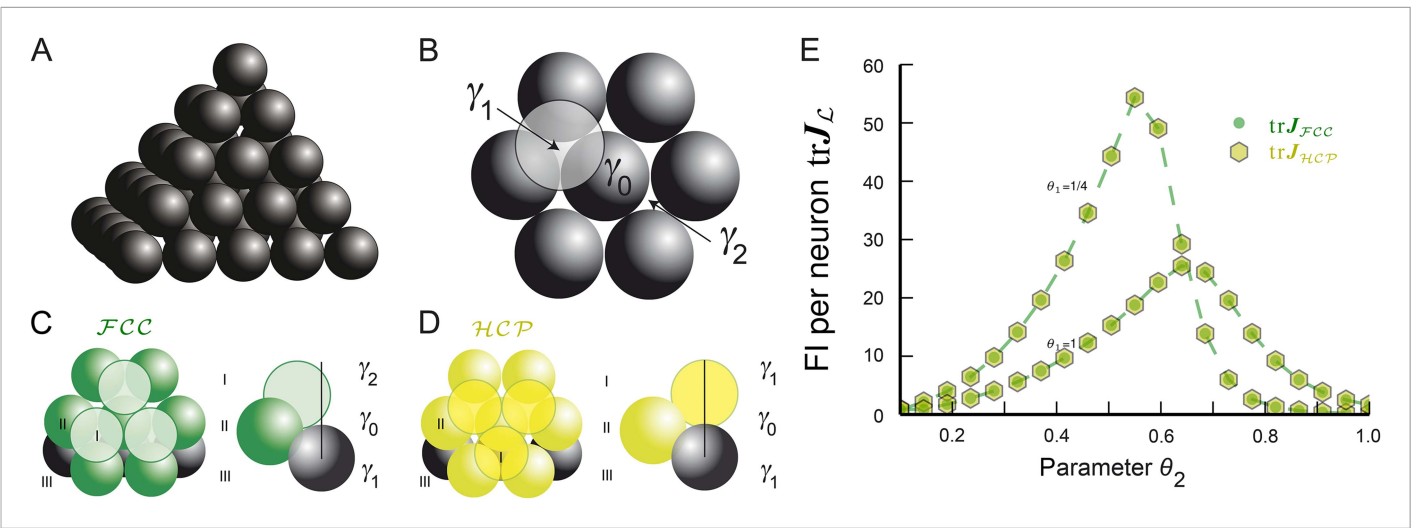

**Figure 5**. Lattice and non-lattice solutions in 3D. (**A**) Stacking of spheres as in an $\mathcal{FCC}$ lattice. In this densest lattice in 3D, each sphere touches 12 other spheres and there are four different planar hexagonal lattices through each node. (**B**) Over a layer of hexagonally arranged spheres centered at $\gamma_0$ (in black) one can put another hexagonal layer by starting from one of six locations, two of which are highlighted, $\gamma_1$ and $\gamma_2$. (**C**) If one arranges the hexagonal layers according to the sequence ($...,\gamma_1, \gamma_0, \gamma_2,...$) one obtains the $\mathcal{FCC}$. Note that spheres in layer I are not aligned with those in layer III. (**D**) Arranging the hexagonal layers following the sequence ($...,\gamma_0, \gamma_1, \gamma_0,...$) leads to the hexagonal close packing $\mathcal{HCP}$. Again, each sphere touches 12 other spheres. However, there is only one plane through each node for which the arrangement of the centers of the spheres is a regular hexagonal lattice. This packing has the same packing ratio as the $\mathcal{FCC}$, but is not a lattice. (**E**) $\mathrm{tr}J_{\mathcal{L}}$ for bump-function $\Omega$ with $\mathcal{L} = \mathcal{FCC}$ and $\mathcal{HCP}$ for various parameter combinations $\theta_1$ and $\theta_2$; $\theta_1$ modulates the decay and $\theta_2$ the support. The two packings have the same packing ratio and for this tuning curve also provide identical spatial resolution. FI: Fisher information.

three vectors, it does not define a lattice (see *Figure 5D*), even though it is as tightly packed as the $\mathcal{FCC}$. Such packings, defined as an arrangement of equal non-overlapping balls (*Conway and Sloane, 1992*; *Hales, 2012*), generalize lattices.

While one can define a grid module for *any* lattice, as we showed above, one *cannot* define a grid module in a meaningful way for an arbitrary packing, due to the lack of symmetry. But for any given packing $\mathcal{P}$ of $\mathbb{R}^D$ by balls $B_1$ of radius 1, one can define a 'grid cell' by generalizing the definition given for lattices (*Equation 7*). To this end, consider the Voronoi partition of $\mathbb{R}^D$ by $\mathcal{P}$. For each location $x \in \mathbb{R}^D$ there is a unique Voronoi cell $V_p$ with node $p \in \mathcal{P}$. One defines the grid cell's tuning curve $\Omega^{\mathcal{P}}(x)$ by assigning the firing rate according to $\Omega(\|p-x\|^2)$ for tuning shape $\Omega$ and distance $\|p-x\|$. Depending on the specific packing, this tuning curve $\Omega^{\mathcal{P}}$ may or may not be periodic. Because a packing $\mathcal{P}$ often has fewer symmetries than a lattice $\mathcal{L}$, the 'grid cells' in an arbitrary $\mathcal{P}$ cannot generally be used to define a 'grid module'. To explain why, consider an arbitrary packing and the unique Voronoi cell $V_0$ that contains the point 0. Choose $M$ uniformly distributed phases $c_1, \ldots, c_M$ within $V_0$. Locations within $V_0$ will then be uniformly covered by shifted tuning curves $\Omega_i(x) := \Omega^{\mathcal{P}}(x-c_i)$. However, typically the different Voronoi cells will neither be congruent, nor have similar volumes. Thus, the $\Omega_i$ will typically *not* cover each Voronoi cell with the same density and will therefore fail to define a proper grid module. This problem does not exist for lattices. Here, the equivalence classes $c_i + \mathcal{L}$ cover each cell with the same density.

Highly symmetric packings, on the other hand, do permit the definition of grid modules. For example, the hexagonal close packing $\mathcal{HCP}$ can be used to define a grid cell $\Omega^{\mathcal{HCP}}(x)$. Using the same symmetry argument from *Equations 9–11*, implies for the FI:

$$\mathbf{J}_{(\Omega,\rho,\mathcal{HCP})}(x) \approx \mathbf{J}_{(\Omega,\rho,\mathcal{HCP})}(0) \approx \frac{M}{\text{vol}(V_0)} \int_{V_0} \mathbf{J}_{\Omega^{\mathcal{HCP}}}(c)\,dc. \tag{18}$$

The maximal in-radius $R$ for the $\mathcal{HCP}$ with grid size $\lambda = 1$ is equal to 1/2. Like for lattices, we assume that $\text{supp}(\Omega) = [0, R]$ and $B_R(0) \subset V_0$. Then the integrand vanishes for distances larger than 1/2 from 0. Hence, we obtain:

$$\mathbf{J}_{(\Omega,\rho,\mathcal{HCP})}(0) \approx \frac{M}{\text{vol}(V_0)} \int_{B_{1/2}(0)} \mathbf{J}_{\Omega^{\mathcal{HCP}}}(c)\,dc. \tag{19}$$

Considering the same tuning shape $\Omega$ and number of phases $M$ for an $\mathcal{FCC}$ lattice, which also has maximal in-radius 1/2, *Equation 13* gives us the following expression for the $\mathcal{FCC}$ lattice:

$$\mathbf{J}_{(\Omega,\rho,\mathcal{FCC})}(0) \approx \frac{M}{\det(\mathcal{FCC})} \int_{B_{1/2}(0)} \mathbf{J}_{\Omega^{\mathcal{FCC}}}(c)\,dc. \tag{20}$$

Since both fundamental domains have the same volumes, that is, $\det(\mathcal{FCC}) = \text{vol}(V_0)$, and the integrands restricted to these balls are identical, that is, $J_{\Omega^{\mathcal{FCC}}}\big|_{B_{1/2}(0)} = J_{\Omega^{\mathcal{HCP}}}\big|_{B_{1/2}(0)}$, we can conclude that grid modules comprising $\mathcal{FCC}$ or $\mathcal{HCP}$-like symmetries have the same FI. We also numerically calculate the trace of the average FI for a module of $\mathcal{HCP}$ grid cells and compare it to the $\mathcal{FCC}$ case. For bump-like tuning curves $\Omega$, both FIs are identical (*Figure 5E*) as expected from the radial symmetry of $\Omega$. As a consequence, grid cells defined by either $\mathcal{HCP}$ or $\mathcal{FCC}$ symmetries provide optimal resolution.

*Figure 5D,E* shows that the cyclic sequences $(\gamma_0, \gamma_1)$ and $(\gamma_1, \gamma_0, \gamma_2)$ lead to $\mathcal{HCP}$ and $\mathcal{FCC}$, respectively. The centers $\gamma_0$, $\gamma_1$, and $\gamma_2$ can also be used to make a final point on packings: there are infinitely many distinct packings with the same density $\pi/(3\sqrt{2})$. They can be constructed by inequivalent words, generated by finitewalks through the triangle with letters $\gamma_0$, $\gamma_1$, and $\gamma_2$ (*Hales, 2012*), with each letter representing one of three orientations for the layers. For instance, $(\gamma_0, \gamma_1, \gamma_0, \gamma_2)$ describes another packing with the same density. All packings share one feature: around each sphere there are exactly 12 spheres, arranged in either $\mathcal{HCP}$ or $\mathcal{FCC}$ lattice fashion (*Hales, 2012*). These packings can also be used to define a grid module, because the density of phases will be uniform in all cells. Furthermore, as in the calculation of the FI for the $\mathcal{HCP}$ and $\mathcal{FCC}$ (*Equation 18–20*) only local integration was necessary, such mixed packings will have equally large, uniform FI as the pure $\mathcal{HCP}$ or $\mathcal{FCC}$ packings.

Only in recent years has it been proven that no other arrangement has a higher packing ratio than the $\mathcal{FCC}$, a problem known as Kepler's conjecture (*Hales, 2005*, *2012*). Based on these results and

our comparison of tr$J_{\mathcal{HCP}}$ and tr$J_{\mathcal{FCC}}$ (*Figure 5E*), we predict that 3D grid cells will correspond to one of these packings. While there are equally dense packings as the densest lattice in 3D, this is not the case in 2D. Thue proved that the hexagonal lattice is unique in being the densest amongst all planar packings (*Thue, 1910*); grid cells in 2D should possess a hexagonal lattice structure.

## Discussion

Grid cells are active when an animal is near one of any number of multiple locations that correspond to the vertices of a planar hexagonal lattice (*Hafting et al., 2005*). We generalize the notion of a *grid cell* to arbitrary dimensions, such that a grid cell's stochastic activity is modulated in a spatially periodic manner within $\mathbb{R}^D$. The periodicity is captured by the symmetry group of the underlying lattice $\mathcal{L}$. A *grid module* consists of multiple cells with equal spatial period but different spatial phases. Using information theory, we then asked which lattice offers the highest spatial resolution.

We find that the resolution of a grid module is related to the packing ratio of $\mathcal{L}$—the lattice with highest packing ratio corresponds to the grid module with highest resolution. Well-known results from mathematics (*Lagrange, 1773*; *Gauss, 1831*; *Conway and Sloane, 1992*) then show that the hexagonal lattice is optimal for representing 2D, whereas the $\mathcal{FCC}$ lattice is optimal for 3D. In 3D, but not in 2D, there are also non-lattice packings with the same resolution as the densest lattice (*Thue, 1910*; *Hales, 2012*). A common feature of these highly symmetric optimal solutions in 3D is that each grid field is surrounded by 12 other grid fields, arranged in either $\mathcal{FCC}$ lattice or hexagonal close packing fashion. These solutions emerge from the set of all possible packings simply by maximizing the resolution, as we showed. However, resolution alone, as measured by the FI, does not distinguish between optimal packing solutions with different symmetries. Whether a realistic neuronal decoder, such as one based on population vector averages, favors one particular solution is an interesting open question.

As we have demonstrated, using the FI makes finding the optimal $\mathcal{L}$ analytically tractable for all dimensions $D$ and singles out densest lattices as optimal tuning shapes under assumptions that are restrictive, but are consistent with experimental measurements (*Hafting et al., 2005*; *Brun et al., 2008*; *Giocomo et al., 2011*). The assumption that the tuning curves must have finite support within the fundamental domain of the lattice corresponds to grid cells being silent outside of the firing field. Indeed, our numerical simulations also showed that for broader tuning curves, grid modules with quadratic lattices can provide more FI than the hexagonal lattice (*Figure 3A*, $\theta_2 \approx 0.6$ and $\theta_1 = 1/4$) and that grid cells with a $\mathcal{C}$ or $\mathcal{BCC}$ lattice can provide more FI than the $\mathcal{FCC}$ (*Figure 4B*, $\theta_2 > 0.65$ and $\theta_1 = 1/4$). For the planar case, *Guanella and Verschure (2007)* show numerically that triangular tessellations yield lower reconstruction errors under maximum-likelihood decoding than equivalently scaled square grids. Complementing this numerical analysis, *Wei et al. (2013)* provide a mathematical argument that hexagonal grids are optimal. To do so, they define the spatial resolution of a single module representing 2D space as the ratio $R = (\lambda/l)^2$, where $\lambda$ is the grid scale and $l$ is the diameter of the circle in which one can determine the animal's location with certainty. For a fixed resolution $R$, the number of neurons required is $N = d \sin(\varphi) R$ in their analysis, where $d$ is the number of tuning curves covering each point in space. As $\varphi \in [\pi/3, \pi/2]$ for the lattice to have unitary length (*Figure 3B*), minimizing $N$ for a fixed resolution $R$ yields $\varphi = \pi/3$; thus, hexagonal lattices should be optimal. Furthermore, Wei et al. show that this result also holds when considering a Bayesian decoder (*Wei et al., 2013*). While Wei et al. minimize $N$ for fixed $l$, we minimize $l$ (in their notation). Like Wei et al., we assume that the tuning curve $\Omega$ is isotropic (notwithstanding the fact that the lattice has preferred directions); unlike these authors, we show that there are conditions under which the firing fields should be arranged in a square lattice, and not hexagonally.

Using the FI gives a theoretical bound for the local resolution of any unbiased estimator (*Lehmann, 1998*). In particular, this local resolution does not take into account the ambiguity introduced by the periodic nature of the lattice. Our analysis is restricted to resolving the animal's position within the fundamental domain. For large neuron numbers $N$ and expected peak spike counts $f_{max}\tau$ the resolution of asymptotically efficient decoders, like the maximum likelihood decoder, or the minimum mean square estimator, can indeed attain the resolution bound given by the FI (*Seung and Sompolinsky, 1993*; *Bethge et al., 2002*; *Mathis et al., 2013*). Thus, for these decoders and conditions the results hold. In contrast, for small neuron numbers and peak spike counts, the optimal codes could be different, just as it has been shown in the past that the optimal tuning width in these

cases cannot be predicted by the FI (*Bethge et al., 2002*; *Yaeli et al., 2010*; *Berens et al., 2011*; *Mathis et al., 2012*).

Maximizing the resolution explains the observed hexagonal patterns of grid cells in 2D, and predicts an $\mathcal{FCC}$ lattice (or equivalent packing) for grid-cell tuning curves of mammals that can freely explore the 3D nature of their environment. Quantitatively, we demonstrated that these optimal populations provide 15.5% (2D) and about 41% (3D) more resolution than grid codes with quadratic or cubic grid cells for the same number of neurons. Although better, this might not seem substantial, at least not at the level of a single grid module. However, as medial entorhinal cortex harbors a nested grid code with at least 5 and potentially 10 or more modules (*Stensola et al., 2012*), this translates into a much larger gain of $1.155^{5 \cdots 10} \approx 2.1 \dots 4.2$ and $\sqrt{2}^{5 \cdots 10} \approx 5.7 \dots 32$, respectively (*Mathis et al., 2012a*, *2012b*). Because aligned grid-cell lattices with perfectly periodic tuning curves imply that the posterior is periodic too (compare *Equation 8*), information from different scales would have to be combined to yield an unambiguous read-out. Whether the nested scales are indeed read out in this way in the brain remains to be seen (*Mathis et al., 2012a*, *2012b*; *Wei et al., 2013*). An alternative hypothesis, as first suggested by Hafting et al., is that the slight, but apparently persistent irregularities in the firing fields across space (*Hafting et al., 2005*; *Krupic et al., 2015*; *Stensola et al., 2015*) are being used. Future experiments should tackle this key question.

We considered perfectly periodic structures (lattices) and asked which ones provide most resolution. However, the first recordings of grid cells already showed that the fields are not exactly hexagonally arranged and that different fields might have different peak firing rates (*Hafting et al., 2005*). More recently, deviations from hexagonal symmetry have gained considerable attention (*Derdikman et al., 2009*; *Krupic et al., 2013*, *2015*; *Stensola et al., 2015*). Such 'defects' modulate the periodicity of the tuning and consequently affect the symmetry of the likelihood function. This might imply that a potential decoder might be able to distinguish different unit cells even given a single module, which is not possible for perfectly periodic tuning curves (compare *Equation 8*). The local resolution, on the other hand, is robust to small, incoherent variations as the FI is a statistical average over many tuning curves with different spatial phases. At a given location, *Equation 9* becomes

$$\boldsymbol{J}_{\varsigma}(x) = \sum_{i=1}^{M} \boldsymbol{J}_{\Omega_i^{\mathcal{L}}}(x - c_i) \approx \int_{x-L} \boldsymbol{J}_{\overline{\Omega^{\mathcal{L}}}}(c)\rho(c)\mathrm{d}c,$$

where $\overline{\Omega^{\mathcal{L}}}$ is the average of the variable tuning curves $\Omega_i^{\mathcal{L}}$. Small variations in the peak rate and grid fields will therefore average out, unless these variations are coherent across grid cells. Thus, resolution bounded by the FI is robust with respect to minor differences in peak firing rates and hexagonality. Similar arguments hold in higher dimensions.

In this study, we focused on optimizing grid modules for an isotropic and homogeneous space, which means that the resolution should be equal everywhere and in each direction of space. From a mathematical point of view, this is the most general setting, but it is certainly not the only imaginable scenario; future studies should shed light on other geometries. Indeed, the topology of natural habitats, such as burrows or caves, can be highly complicated. Higher resolution might be required at spatial locations of behavioral relevance. Neural representations of 3D space may also be composed of multiple 1D and 2D patches (*Jeffery et al., 2013*). However, the mere fact that these habitats involve complicated low-dimensional geometries does not imply that an animal cannot acquire a general map for the environment. Poincaré already suggested that an isotropic and homogeneous representation for space can emerge out of non-Euclidean perceptual spaces, as one can move through physical space by learning the motion group (*Poincaré, 1913*). An isotropic and homogeneous representation of 3D space facilitates (mental) rotations in 3D and yields local coordinates that are independent of the environment's topology. On the other hand, the efficient-coding hypothesis (*Barlow, 1959*; *Atick, 1992*; *Simoncelli and Olshausen, 2001*) would argue that surface-bound animals might not need to dedicate their limited neuronal resources to acquiring a full representation of space, as flying animals might have to do, so that representations of 3D space will be species-specific (*Las and Ulanovsky, 2014*). Desert ants represent space only as a projection to flat space (*Wohlgemuth et al., 2001*; *Grah et al., 2007*). Likewise, experimental evidence suggests that rats do not encode 3D space in an isotropic manner (*Hayman et al., 2011*), but this might be a consequence of the specific anisotropic spatial navigation tasks these rats had to perform. Data from

flying bats, on the other hand, demonstrate that, at least in this species, place cells represent 3D space in a uniform and nearly isotropic manner (*Yartsev and Ulanovsky, 2013*). The 3D, toroidal head-direction system in bats also suggests that they have access to the full motion group (*Finkelstein et al., 2014*). Our theoretical analysis assumes that the same is true for bat grid cells and that they have radially symmetric firing fields. From these assumptions, we showed the grid cells' firing fields should be arranged on an $\mathcal{FCC}$ lattice or packed as $\mathcal{HCP}$. Interestingly, such solutions also evolve dynamically in a self-organizing network model for 3D (*Stella et al., 2013*; *Stella and Treves, 2015*) that extends a previous 2D system which exhibits hexagonal grid patterns (*Kropff and Treves, 2008*). Experimentally, the effect of the arena's geometry on grid cells' tuning and anchoring has also been a question of great interest (*Derdikman et al., 2009*; *Krupic et al., 2013*, *2015*; *Stensola et al., 2015*). First, let us note that even though the environment might be finite, the grid-cell representation need not be constrained by it. In particular, the firing fields are not required to be contained within the confines of the four walls of a box—experimental observations show that walls can intersect the firing fields (so that one measures only a part of the firing field). On the other hand, the borders clearly distort the hexagonal arrangement of nearby firing fields in 2D environments (*Stensola et al., 2015*), whereas central fields are more perfectly arranged. Deviations are also observed when only a few fields are present in the arena (*Krupic et al., 2015*). One might expect similar deviations in 3D, such as for bats flying in a confined space. Our mathematical results rely on symmetry arguments that do not cover non-periodic tuning curves. Given that the resolution is related to the packing ratio of a lattice, extensions of the theory to general packings might allow one to draw on the rich field of optimal finite packings (*Böröczky, 2004*; *Toth et al., 2004*), thereby providing new hypotheses to test.

Many spatially modulated cells in rat medial entorhinal cortex have hexagonal tuning curves, but some have firing fields that are spatially periodic bands (*Krupic et al., 2012*). The orientation of these bands tends to coincide with one of the lattice vectors of the grid cells (as the lattices for different grid cells share a common orientation), so band cells might be a layout 'defect'. In this context, we should point out that the lattice solutions are not globally optimal. For instance, in 2D, a higher resolution can result from two systems of nested 1D grid codes, which are aligned to the $x$ and $y$ axis, respectively, than from a lattice solution with the same number of neurons. The 1D cells would behave like band cells (*Krupic et al., 2012*). Similar counterexamples can be given in higher dimensions too. The anisotropy of the spatial tuning in grid cells of climbing rats when encoding 3D space (*Hayman et al., 2011*) might capitalize on this gain (*Jeffery et al., 2013*). Radial symmetry of the tuning curve may also be non-optimal. For example, two sets of elliptically tuned 2D unimodal cells, with orthogonal short axes, typically outperform unimodal cells with radially symmetric tuning curves (Wilke and Eurich, 2002). Why experimentally observed place fields and other tuning curves seem to be isotropically tuned is an open question (*O'Keefe and Dostrovsky, 1971*; *Yartsev and Ulanovsky, 2013*).

Grid cells which represent the position of an animal (*Hafting et al., 2005*) have been discovered only recently. By comparison, in technical systems, it has been known since the 1950s that the optimal quantizers for 2D signals rely on hexagonal lattices (*Gray and Neuhoff, 1998*). In this context, we note that lattice codes are also ideally suited to cover spaces that involve sensory or cognitive variables other than location. In higher-dimensional feature spaces, the potential gain could be enormous. For instance, the optimal eight-dimensional (8D) lattice is about 16 times denser than the orthogonal 8D lattice (*Conway and Sloane, 1992*) and would, therefore, dramatically increase the resolution of the corresponding population code. Advances in experimental techniques, which allow one to simultaneously record from large numbers of neurons (*Ahrens et al., 2013*; *Deisseroth and Schnitzer, 2013*) and to automate stimulus delivery for dense parametric mapping (*Brincat and Connor, 2004*), now pave the way to search for such representations in cortex. For instance, by parameterizing 19 metric features of cartoon faces, such as hair length, iris size, or eye size, Freiwald et al. showed that face-selective cells are broadly tuned to multiple feature dimensions (*Freiwald et al., 2009*). Especially in higher cortical areas, such joint feature spaces should be the norm rather than the exception (*Rigotti et al., 2013*). While no evidence for lattice codes was found in the specific case of face-selective cells, data sets like this one will be the test-bed for checking the hypothesis that other nested grid-like neural representations exist in cortex.

## Materials and methods

We study population codes of neurons encoding the $D$-dimensional space by considering the FI $J$ as a measure for their resolution. The population coding model, the construction to periodify a tuning shape $\Omega$ onto a lattice $\mathcal{L}$ with center density $\rho$, as well as the definition of the FI, are given in the main text. In this section we give further background on the methods.

### Scaling of grid cells and the effect on $J_\varsigma$

How is the resolution of a grid module affected by dilations? Let us assume we have a grid module with signature $\varsigma = (\Omega, \rho, \mathcal{L})$, as defined in the main text, and that $\lambda > 0$ is a scaling factor. Then $\lambda\varsigma := (\Omega(\lambda r), \rho(\lambda x), \lambda \cdot \mathcal{L})$ is a grid module too, and the corresponding tuning curve $(\Omega \circ \lambda)_{\lambda\mathcal{L}}$ satisfies:

$$(\Omega \circ \lambda)_{\lambda\mathcal{L}}(x) = \Omega_{\mathcal{L}}(\lambda x). \tag{21}$$

Thus, the tuning curve $(\Omega \circ \lambda)_{\lambda\mathcal{L}}$ is a scaled version of $\Omega_{\mathcal{L}}$. What is the relation between the FI of the initial grid module and the rescaled version? Let us fix the notation: $\rho(c) = \sum_i^N \delta(c - c_i)$. From the definition of the population information (*Equation 9*), we calculate

$$J_{\lambda\varsigma}(0) = \sum_{i=1}^{M} J_{(\Omega \circ \lambda)_{\lambda\mathcal{L}}}(\lambda c_i) = \sum_{i=1}^{M} J_{\Omega_{\mathcal{L}}}(c_i) \cdot \frac{1}{\lambda^2} = \frac{1}{\lambda^2} J_\varsigma(0), \tag{22}$$

where in the second step we used the re-parameterization formula of the FI (*Lehmann, 1998*). This shows that the FI of a grid module scaled by a factor $\lambda$ is the same as the FI of the initial grid module times $1/\lambda^2$.

### Population FI for Poisson noise with radially symmetric tuning

In the 'Results' section, we give a concrete example for Poisson noise and the bump function. Here we give the necessary background. *Equation 13* states that

$$J_\varsigma(0) \approx \frac{M}{\det(\mathcal{L})} \int_{B_R(0)} J_{\Omega^{\mathcal{L}}}(c) dc.$$

One would like to know $\int_{B_R(0)} J_{\Omega^{\mathcal{L}}}(c) dc$ for various tuning shapes $\Omega$ with $\mathrm{supp}(\Omega) \leq R$. Consider $x \in L$ and $\alpha \in \{1, \ldots, D\}$. Then:

$$\frac{\partial \ln P(K|x)}{\partial x_\alpha} = \frac{\partial \ln P(K, s)}{\partial s}\bigg|_{s = \Omega^{\mathcal{L}}(x)} \cdot \Omega'\left(||x||^2\right) f_{max}\tau \, 2x_\alpha. \tag{23}$$

Together with the definition of the FI *Equation 13*, this yields

$$J_{\Omega^{\mathcal{L}}}(x)_{\alpha\beta} = 4x_\alpha x_\beta f_{max}^2 \tau^2 \Omega'\left(||x||^2\right)^2 \cdot \underbrace{\sum_K \left(\frac{\partial}{\partial s}\ln P(K, s)|_{s = \Omega^{\mathcal{L}}(x)}\right)^2 \cdot P(K, \Omega^{\mathcal{L}}(x))}_{=:\mathcal{N}\left(||x||^2\right)}. \tag{24}$$

Note that for $\alpha \neq \beta$ this function is odd in $x$. Thus, when averaging these individual contributions over a symmetric fundamental domain $L$: $\int_L J_{\Omega^{\mathcal{L}}}(c)_{\alpha\beta} dc = 0$ for $\alpha \neq \beta$. Thus, the diagonal entries are all identical. This also holds for any fundamental domain $L$ when $B_R(0) \subset L$, because $B_R(0)$ is symmetric.

For Poisson spiking $\mathcal{N}(||c||^2)$ has a particularly simple form, namely $\mathcal{N}(||c||^2) = 1/(f_{max}\tau \, \Omega(||c||^2))$. The trace of the FI matrix becomes

$$\mathrm{tr}J_\varsigma(0) = 4f_{max}\tau \int_{B_R(0)} \underbrace{||c||^2 \frac{\Omega'\left(||c||^2\right)^2}{\Omega\left(||c||^2\right)}}_{=:\mathcal{F}(c)} dc. \tag{25}$$

Thus, the trace only depends on the tuning shape $\Omega$ and its first derivative. In the main text, we use the following specific tuning shape:

$$\Omega(r) = \begin{cases} \exp\left(\dfrac{\theta_1}{\theta_2^2} - \dfrac{\theta_1}{\theta_2^2 - r^2}\right) & \text{if } |r| < \theta_2 \\ 0 & \text{otherwise} \end{cases}. \tag{26}$$

This type of function is often called 'bump function' in topology, as it has a compact support but is everywhere smooth (i.e., infinitely times continuously differentiable). In particular, the support of this function is $[0, \theta_2)$, and is therefore controlled by the parameter $\theta_2$. The other parameter $\theta_1$ controls the slope of the bump's flanks (see upper panels of *Figure 3—figure supplement 1*).

For the bump-function $\Omega$ and radius $r = \sqrt{\sum_\alpha^D x_\alpha^2}$ the integrand for the FI is given by

$$\mathcal{F}(r) = \begin{cases} \dfrac{4\theta_1^2 r^2}{(\theta_2^2 - r^2)^4} \exp\left(\dfrac{\theta_1}{\theta_2^2} - \dfrac{\theta_1}{\theta_2^2 - r^2}\right) & \text{if } |r| < \theta_2 \\ 0 & \text{otherwise} \end{cases}. \tag{27}$$

The lower panels of *Figure 3—figure supplement 1* depict the integrand of *Equation 25*, defined as $\mathcal{F}(r)$. This function shows how much FI a cell at a particular distance contribute to the location 0. By integrating the FI over the fundamental domain $L$ for a lattice $\mathcal{L}$ one gets $J_\varsigma(0)$, that is, the average FI contributions from all neurons (as shown in *Figures 3, 4, 5E*).

## Acknowledgements

We thank Kenneth Blum and Mackenzie Amoroso for discussions. AM is grateful to Mackenzie Amoroso for graphics help and Ashkan Salamat for discussing the nomenclature in crystallography.

## Additional information

### Funding

| Funder | Grant reference | Author |
| --- | --- | --- |
| Bundesministerium für Bildung und Forschung | 01 GQ 1004A | Martin B Stemmler, Andreas VM Herz |
| Deutsche Forschungsgemeinschaft | MA 6176/1-1 | Alexander Mathis |
| European Commission | PIOF-GA-2013-622943 | Alexander Mathis |

The funders had no role in study design, data collection and interpretation, or the decision to submit the work for publication.

### Author contributions

AM, Conception and design, Analysis and interpretation of data, Drafting or revising the article; MBS, AVMH, Conception and design, Drafting or revising the article

### Author ORCIDs

Alexander Mathis, http://orcid.org/0000-0002-3777-2202

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
