## [Decision Letter]

Thank you for sending your work entitled “Probable nature of higher-dimensional symmetries underlying mammalian grid-cell activity patterns” for consideration at *eLife*. Your article has been favorably evaluated by Eve Marder (Senior editor), Mark Goldman (guest Reviewing editor), and two reviewers.

The Reviewing editor and the reviewers discussed their comments before we reached this decision, and the Reviewing editor has assembled the following comments to help you prepare a revised submission.

In the interesting manuscript “Probable nature of higher-dimensional symmetries underlying mammalian grid-cell activity patterns”, Mathis et al. provide the first principled and theoretically-rooted predictions for how the activity of grid cells might look like in 3D and, more generally, provide interesting predictions on the possible nature of other neural codes for higher-dimensional stimulus spaces, such as e.g. in the prefrontal cortex. The authors do this by considering the Fisher information in a neural code consisting of firing fields that are arranged in a periodic structure in multidimensional space. Under several mathematical idealizations and assumptions, it is argued that the best arrangement for the firing fields is one that leads to maximal packing of the periodic firing fields. These predictions can be tested experimentally in grid-cell recordings in flying bats.

The demonstration that *FCC* packing of grids is an optimal arrangement of firing fields from the coding perspective is an elegant, intriguing, and valuable result that should be of interest to many neuroscientists, biologists, and physicists. However, despite suggestions to the contrary, the work does not provide a formal argument for optimality of other close packing arrangements (such as *HCP* and the non-periodic arrangements) and it is not obvious that such arrangements are equivalent from the Fisher information perspective. Furthermore, the work bears close resemblance to unpublished (but cited by the authors and publically posted) work by Balasubramanian et al.

1) The main mathematical derivations are correct and the argument in favor of maximal packing for *FCC* grids is simple and elegant. However, the work does not provide a formal argument for optimality of other close packing arrangements (such as *HCP* and the non-periodic arrangements) and it is not clear that all the maximal packing arrangements are equivalent in terms of the Fisher information. The derivation leading to [Disp-formula equ12] is precise and clear for periodic arrangements that contain one firing field per unit cell (such as the hexagonal packing in 2 dimensions or *FCC*), but there is a missing link between this derivation and statements about maximal close packing in more general arrangements. This means that the statements on the periodic *HCP* lattices, as well as the non-periodic hybrids between *FCC* and *HCP*, are not clearly justified. This issue, and specifically whether *HCP* is as good as *FCC* (or not), must be addressed before the manuscript can be considered for publication.

2) The authors should provide more discussion of the similarities and differences of the present work from that in 2-D lattices by Balasubramanian's group. The reply to reviewers should include specification of how this work provides a significant conceptual advance.

3) *HCP* and *FCC* are optimal packings for infinite 3D spaces, but, to our knowledge, there is no mathematical proof for what is the optimal sphere packing for a finite-sized 3D space, such as typical laboratory arenas and rooms. The authors should discuss this “finite size problem”, and in particular, what would happen to the blobs of the grid along the walls of the arena/room. For example, can the notion of maximal-FI/optimal-packing explain the distortions of 2D grids that were recently discovered by O'Keefe's group (in a trapezoid arena) and by the Mosers group (shearing of the grids along the wall)? If so, what would be the implications for 3D grid cell activity near the walls? In a similar vein, could there be a scenario (i.e. a certain ratio of room size to sphere radius) where, for a finite-sized room, an optimal packing will in fact yield elongated columnar hexagons (or elongated “strings of spheres”), as was suggested by the study in rats from the Jeffery group (27)?

4) The manuscript contains some inconsistencies and confusing statements in the discussion of the *FCC* and *HCP* lattices. The classification of lattices in 3 dimensions is well established in some areas of mathematics, physics, and engineering, so it is important to be precise and to conform with conventional nomenclature. The *FCC* arrangement is referred to in the manuscript as the only optimal lattice packing in 3 dimensions, and the *HCP* arrangement is referred to as a non-lattice arrangement because it cannot be spanned by three vectors with integer coefficients. Both of these statements disagree with the conventional classification of lattices: the *FCC* as well as the *HCP* arrangements are perfectly periodic structures.

5) The idea that the function of grid cells is to produce a nested code of position is a hypothesis, not only in three dimensional spaces but also in two dimensions, since the role of grid cells in spatial coding is far from being established experimentally, and even the structure of the grid cell code in rodents has been characterized only in very simple and small environments. This does not diminish the relevance of the theory, but we would prefer to see a larger degree of caution in the discussion on the biological reality.

Minor comments:

1) Due to the existence of global ambiguities, the ability to discriminate between very close locations (quantified by the Fisher information) might not be sufficient to achieve high resolution, as addressed in previous works by the same authors as well as other authors. It is not clear to me that local optimization of the Fisher Information is optimal from this broader perspective, even if this seems reasonable. I suggest the authors clarify or at least acknowledge this point.

2) In spaces of dimension 3 and higher it has been argued by Zhang and Sejnowsky (1998), and by Pouget, Deneve, and Ducom (1999) that it is beneficial to have wide tuning curves in order to maximize the Fisher Information. This brings into question the validity of the assumption that firing fields will have compact support in an optimal code in high dimensional spaces. In general, I thought that the assumption of compact support is perfectly legitimate (and conforms with what we know about grid cells in rodents and crawling bats), but it should be emphasized more clearly that the results rely on this assumption.

3) The diagonal elements of the Fisher information matrix are not the appropriate quantity to consider as a tight bound on the resolution. Instead, one has to consider the diagonal elements of the inverse Fisher information matrix. This is a minor issue because the inverse information matrix scales with the volume of the unit cell in agreement with the conclusions of the derivation. Nevertheless I suggest the authors address this comment by modifying the text in the subsection headed “Fisher information of a grid module with lattice *L*” below [Disp-formula equ12].

4) The statement below [Disp-formula equ4] on isotropy is confusing because the grid cell representation is not truly isotropic (there are obviously special directions in space).

5) We failed to identify in the manuscript a powerful link with cryptography or to the role of maximal close packing in coding theory (which appears there in a different context). Therefore, in my opinion, these declarations in the Introduction do not serve a meaningful or useful purpose.

---

## [Author Response]

*1) The main mathematical derivations are correct and the argument in favor of maximal packing for* FCC *grids is simple and elegant. However, the work does not provide a formal argument for optimality of other close packing arrangements (such as* HCP *and the non-periodic arrangements) and it is not clear that all the maximal packing arrangements are equivalent in terms of the Fisher information. The derivation leading to*
[Disp-formula equ12]
*is precise and clear for periodic arrangements that contain one firing field per unit cell (such as the hexagonal packing in 2 dimensions or* FCC*), but there is a missing link between this derivation and statements about maximal close packing in more general arrangements. This means that the statements on the periodic* HCP *lattices, as well as the non-periodic hybrids between* FCC *and* HCP*, are not clearly justified. This issue, and specifically whether* HCP *is as good as* FCC *(or not), must be addressed before the manuscript can be considered for publication*.

We thank the reviewers for this comment. We clarified the argument, which shows the equivalence of the FI of these structures. As this updated part is fairly long, please refer to the subsection headed “Equally optimal non-lattice solutions for grid cell tuning” in the revised manuscript.

*2) The authors should provide more discussion of the similarities and differences of the present work from that in 2-D lattices by Balasubramanian's group. The reply to reviewers should include specification of how this work provides a significant conceptual advance*.

We thank the reviewers for this comment. Guanella and Verschure numerically demonstrated in 2007 that the hexagonal lattice is the optimal regular lattice in planar space with respect to a maximum likelihood decoder and specific tuning curves. Wei et al. showed that a nested grid code’s resolution (defined as range / smallest scale) is highest when the grid pattern is hexagonal. They demonstrated this by analyzing the resolution for two specific decoders (winner takes it all; Bayesian) over different choices of planar lattices.

Our work advances these earlier studies in two ways:

A) We analytically derive that for any radially symmetric tuning shape (Omega in our manuscript), which satisfies the condition that grid fields are well separated, the grid module with the densest pattern provides the highest spatial resolution. We uncover conditions under which the densest arrangement is not optimal; for instance, see Figures 3 and 4 for large *θ*_2_.

B) Our results hold for any number of dimensions of space, making specific predictions for putative grid cells in flying or swimming mammals. In the Ulanovsky laboratory at the Weizmann Institute, Gily Ginosar and colleagues are currently recording from the entorhinal cortex in flying bats (SfN, 2014), and preliminary results are promising.

In the revised manuscript we now more extensively discuss the prior work by Guanella/Verschure and Balasubramanian’s group. Specifically, the citation in the Introduction now reads (second paragraph):

“Theoretical and numerical studies suggest that the hexagonal lattice structure is best suited for representing such a two-dimensional (2D) space [Guanella 2007, [40] and Wei 2013].”

In the Results section (subsection headed “Optimal two-dimensional grid cells”) we added:

“Our theory implies that for radially symmetric tuning curves the hexagonal lattice provides the best resolution […] Guanella and Verschure numerically compared hexagonal to other regular lattices based on maximum likelihood decoding [Guanella 2007].”

In addition, in the Discussion we write:

“[…] For the planar case, Guanella and Verschure [Guanella 2007] show numerically that triangular tessellations yield lower reconstruction errors […]; unlike these authors, we show that there are conditions under which the firing fields should be arranged in a square lattice, and not hexagonally.”

*3)* HCP *and* FCC *are optimal packings for infinite 3D spaces, but, to our knowledge, there is no mathematical proof for what is the optimal sphere packing for a finite-sized 3D space, such as typical laboratory arenas and rooms. The authors should discuss this “finite size problem”, and in particular, what would happen to the blobs of the grid along the walls of the arena/room. For example, can the notion of maximal-FI/optimal-packing explain the distortions of 2D grids that were recently discovered by O'Keefe's group (in a trapezoid arena) and by the Mosers group (shearing of the grids along the wall)? If so, what would be the implications for 3D grid cell activity near the walls? In a similar vein, could there be a scenario (i.e. a certain ratio of room size to sphere radius) where, for a finite-sized room, an optimal packing will in fact yield elongated columnar hexagons (or elongated “strings of spheres”), as was suggested by the study in rats from the Jeffery group (*[27]*)*?

We thank the reviewers for this very interesting and difficult question, but as we highlight in the updated part on grid cells defined according to packings, our theory cannot be easily extended to arbitrary packings. We think that this is beyond the scope of this study. We substantially expanded the Discussion regarding these issues and included references to some of the most important results regarding optimal packings in finite dimensional spaces that we are aware of.

Specifically, we discuss “periodicity” in the following passage:

“We considered perfectly periodic structures (lattices) and asked which ones provide most resolution. […] Thus, resolution bounded by the FI is robust with respect to minor differences in peak firing rates and hexagonality. Similar arguments hold in higher dimensions.”

We also added a section on the environment’s shape:

“Experimentally, the effect of the arena’s geometry on grid cells tuning and anchoring has also been a question of great interest […] extensions of the theory to general packings might allow one to draw on the rich field of optimal finite packings [Toth 2004, [7]], thereby providing new hypotheses to test.”

In the context of geometry we had already discussed [27] (e.g. in the sixth paragraph of the Discussion), but additionally we highlighted this work in the context of axes being more efficient than lattices (in the Discussion): “The anisotropy of the spatial tuning in grid cells of climbing rats when encoding 3D space [Hayman 2011] might capitalize on this gain [Jeffery 2013].”

*4) The manuscript contains some inconsistencies and confusing statements in the discussion of the* FCC *and* HCP *lattices. The classification of lattices in 3 dimensions is well established in some areas of mathematics, physics, and engineering, so it is important to be precise and to conform with conventional nomenclature. The* FCC *arrangement is referred to in the manuscript as the only optimal lattice packing in 3 dimensions, and the* HCP *arrangement is referred to as a non-lattice arrangement because it cannot be spanned by three vectors with integer coefficients. Both of these statements disagree with the conventional classification of lattices: the* FCC *as well as the* HCP *arrangements are perfectly periodic structures*.

There are certain differences in nomenclature across fields. We used a standard reference in mathematics: “Sphere Packings, Lattices and Groups” by J.H. Conway and N.J.A. Sloane.Their nomenclature has the advantage of being unambiguous and proves to be sufficient for the study of grid cell symmetries. A number of references, including Gauss’ proof of the densest lattice, Thue’s paper and Hales’ book on the Kepler conjecture, use the standard definitions from mathematics, rather than physics.

In solid state physics and crystallography (see Whittaker’s book, for instance), a lattice is defined as the imaginary array of points in space that constitutes a motif that repeats itself. The motif for a crystal can contain multiple nodes (spheres); in principle, this motif can be highly complicated.

To clarify this distinction for the reader, we added the following part, after the definition of a grid cell (in the subsection headed “Periodic tuning curves”):

“We will not consider degenerate lattices. In this work, we follow the nomenclature from Conway & Sloane [Conway 1993]. Applied fields might differ slightly in their terminology, especially regarding naming conventions for packings, which are generalizations of lattices [[59], Nelson 2002]. We will address these generalizations of lattices below.”

*5) The idea that the function of grid cells is to produce a nested code of position is a hypothesis, not only in three dimensional spaces but also in two dimensions, since the role of grid cells in spatial coding is far from being established experimentally, and even the structure of the grid cell code in rodents has been characterized only in very simple and small environments. This does not diminish the relevance of the theory, but we would prefer to see a larger degree of caution in the discussion on the biological reality*.

We thank the reviewers for this cautionary reminder. We added such a note to the Discussion. In particular, we wrote:

“Because aligned grid-cell lattices with perfectly periodic tuning curves imply that the posterior is periodic, too (compare [Disp-formula equ8]), information from different scales would have to combined to yield an unambiguous read-out. Whether the nested scales are indeed read out in this way in the brain remains to be seen [Wei 2013, [40] and [40]]. An alternative hypothesis, as first suggested by Hafting et al., is that the slight, but apparently persistent irregularities in the firing fields across space [Hafting 2005, Stensola 2015 and Krupic 2015] are being used. Future experiments should tackle this key question.”

Minor comments:

1) Due to the existence of global ambiguities, the ability to discriminate between very close locations (quantified by the Fisher information) might not be sufficient to achieve high resolution, as addressed in previous works by the same authors as well as other authors. It is not clear to me that local optimization of the Fisher Information is optimal from this broader perspective, even if this seems reasonable. I suggest the authors clarify or at least acknowledge this point.

We thank the reviewers for this comment. We wanted to bring the following extended paragraph (in the Discussion section) to the attention of the reviewers, where we acknowledge this issue:

“Using the FI gives a theoretical bound for the local resolution of any unbiased estimator [Lehmann 1998]. […] for small neuron numbers and peak spike counts, the optimal codes could be different, just as it has been shown in the past that the optimal tuning width in these cases cannot be predicted by the FI [Bethge 2001, [40], Yaeli 2010 and Berens 2011].”

2) In spaces of dimension 3 and higher it has been argued by Zhang and Sejnowsky (1998), and by Pouget, Deneve, and Ducom (1999) that it is beneficial to have wide tuning curves in order to maximize the Fisher Information. This brings into question the validity of the assumption that firing fields will have compact support in an optimal code in high dimensional spaces. In general, I thought that the assumption of compact support is perfectly legitimate (and conforms with what we know about grid cells in rodents and crawling bats), but it should be emphasized more clearly that the results rely on this assumption.

This is an important point. While the papers mentioned above consider unimodal tuning curves (that might represent place fields, for instance), we treat periodic tuning curves. The construction in [Disp-formula equ7] of the text implies that as the support of the tuning curve becomes broad, the firing rate no longer dips to zero on the border *∂L* of the fundamental domain. Eventually, for very broad tuning, the tuning curve becomes essentially flat and ceases to be ‘informative’, even though the support of Ω(*r*) remains finite.

The fractional volume of *L* outside *B*_*R*_(0),

det(*L*)
*−*
vol(*B*_*R*_(0))

vol(*B*_*R*_(0))

can increase for certain *L* as *D* increases. Therefore, wider tuning curves might offer an increasing advantage in higher-dimensional spaces, as they support the volume contribution to the Fisher Information from outside the ball *B*_*R*_(0). But even for *D* = 2 and *D* = 3, satisfying the condition supp(Ω) = [0*, R*] with *B*_*R*_(0) ⊂ *L* is not optimal with respect to the Fisher information for any tuning shape, as seen in Figures 3 and 4. In the revised manuscript, we highlight this fact (in the Discussion). In the Results section, we use a tuning curve Ω(*r*) that is a bump function with two parameters *θ*_1_ and *θ*_2_, where *θ*_2_ is the radius of supp(Ω). If one were to optimize both parameters, one would recover the condition supp(Ω) = [0*, R*], where specifically *R* = 1*/*2 is the maximal in-radius. The Fisher information is maximized when *θ*_1_
*«* 1, as long as the number of neurons *M* is large. For *θ*_1_
*«* 1, most of the Fisher information results from a radial band close to the support radius *θ*_2_; indeed, in the limit *θ*1 *→* 0, the bump function becomes a step function. Figure 6 shows that then indeed for both 2D and 3D the optimal tuning width satisfies the condition that supp(Ω) = [0*,* 1*/*2].

Author response image 1.Average trace tr***J****L* of FI for uniformly distributed grid cells Ω^*L*^ for Poisson noise and bump tuning shape Ω. Left Panel: tr*J*_*L*_ for hexagonal (*H* in green) and square (*Q* in blue) lattices are shown for different *θ*_2_ values, with *θ*_1_ varying from 0*.*25 (lowest pair of lines), 0*.*1 (middle pair of lines) to 0*.*01 (top pair of lines). Thus, with decreasing *θ*_1_ the FI grows and reaches the maximum at the in-radius of both lattices *θ*_2_ = 0*.*5. Right panel: tr*J*_*L*_ for face-centered cubic (*FCC* in green), body-centered cubic (*BCC* in red) and cubic (*C* in blue) are shown for various *θ*_2_ and *θ*_1_ varying from 0*.*25 (lowest triple of lines), 0*.*1 (middle triple of lines) to 0*.*01 (top triple of lines). Again, the FI of the smallest *θ*_1_ is best and reaches its peak at *θ*_2_ = 0*.*5.**DOI:**
http://dx.doi.org/10.7554/eLife.05979.012

To take these considerations into account, we extended the paragraph about the tuning width to:

“The condition supp(Ω) = [0, *R*] with *B*_*R*_(0) ⊂ *L*, although restrictive, is consistent with experimental observations that grid cells tend to stop firing between grid fields […] the densest lattice provides the highest resolution for any tuning shape Ω, as we just demonstrated.”

*3) The diagonal elements of the Fisher information matrix are not the appropriate quantity to consider as a tight bound on the resolution. Instead, one has to consider the diagonal elements of the inverse Fisher information matrix. This is a minor issue because the inverse information matrix scales with the volume of the unit cell in agreement with the conclusions of the derivation. Nevertheless I suggest the authors address this comment by modifying the text in the subsection headed “Fisher information of a grid module with lattice* L*” below*
[Disp-formula equ12].

This is absolutely correct. We changed the sentence to:

“The error for each coordinate axis is thus bounded by the same value, i.e. the inverse of the diagonal element 1*/****J***_*ς*_ (0)_*ii*_, for such a population.”

4) The statement below [Disp-formula equ4] on isotropy is confusing because the grid cell representation is not truly isotropic (there are obviously special directions in space).

We are sorry for the confusion. With isotropy we are not referring to the tuning curves’ symmetries (which are indeed not isometric), but rather to the posterior’s. As we calculated in the manuscript, the Fisher information for any lattice (as long as the support of the tuning curve is contained in the fundamental domain and the neuron number *M* is sufficiently large) is a diagonal matrix with identical entries. This means that the bound given by the Fisher information is Gaussian independent of the symmetries of the lattice. Similarly, all typical errors that, for instance, a Maximum likelihood decoder makes are radially symmetrically distributed around the true value.

We added the following part to the manuscript (subsection headed “Resolution and Fisher information”):

“These two conditions assure that the population has the same resolution at any location and along any spatial axis […]but the posterior is radially symmetric around any given location for a module of such grid cells.”

5) We failed to identify in the manuscript a powerful link with cryptography or to the role of maximal close packing in coding theory (which appears there in a different context). Therefore, in my opinion, these declarations in the Introduction do not serve a meaningful or useful purpose.

We thank the reviewers for this comment. We dropped this declaration regarding the link.

We also agree with the reviewers that sphere packings and coverings appear in various forms across cryptography and coding theory. Rather than claiming the existence of a practical link, we wanted to highlight that the problem of sphere packings underlies both the design of optimal codes in coding theory, for instance for the Gaussian white noise channel (Compare to Conway / Sloane), and provides the answer to the neuronal coding problem we considered.

Thus, we would like to suggest to add the relative clause: “which also plays an important role in other coding problems and cryptography [[50], Conway 1992 and Gray 1998]” after the following sentence in the Introduction: “Even though the firing fields between cells overlap, so as to ensure uniform coverage of space, we show how resolving the population’s Fisher information can be mapped onto the problem of packing non-overlapping spheres, which also plays an important role in other coding problems and cryptography [[50], Conway 1992 and Gray 1998].”